# Lego-Edit: A General Image Editing Framework with Model-Level Bricks and MLLM Builder

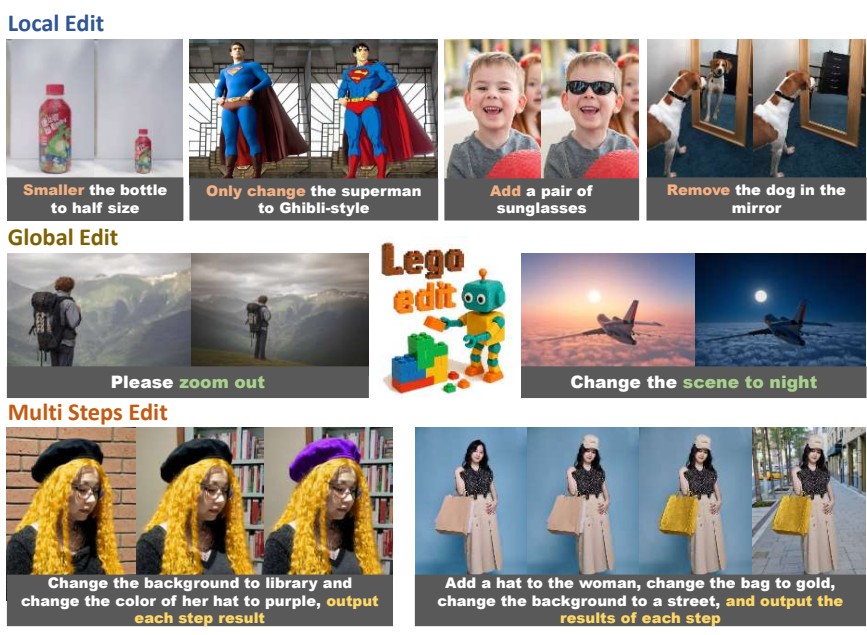

Figure 1: Editing examples of Lego-Edit.

## Abstract

Instruction-based image editing has garnered significant attention due to its direct interaction with users. However, real-world user instructions are immensely diverse, and existing methods often fail to generalize effectively to instructions outside their training domain, limiting their practical application. To address this, we propose Lego-Edit, which leverages the generalization capability of Multimodal Large Language Model (MLLM) to organize a suite of model-level editing tools to tackle this challenge. Lego-Edit incorporates two key designs: (1) a model-level toolkit comprising diverse models efficiently trained on limited data and several image manipulation functions, enabling fine-grained composition of editing actions by the MLLM; and (2) a three-stage progressive reinforcement learning approach that uses feedback on unannotated, open-domain instructions to train the MLLM, equipping it with generalized reasoning capabilities for handling real-world instructions. Experiments demonstrate that Lego-Edit achieves state-of-the-art performance on GEdit-Bench and ImgBench. It exhibits robust reasoning capabilities for open-domain instructions and can utilize newly introduced editing tools without additional fine-tuning.

## 1 INTRODUCTION

Instruction-based image editing methods accept natural language instructions as input and modify the input image accordingly. It enables intuitive human-computer interaction, offering broad application potential. However, the significant diversity inherent in real-world editing instructions poses a substantial challenge for image editing systems in handling flexible user commands.

Existing approaches for instruction-based editing are broadly classified into two categories. End-to-end methods (Labs, 2024a; Liu et al., 2025) train a single generative model to learn both instruction comprehension and pixel mapping for editing implicitly. As shown in Fig. 2 (a), these methods are primarily constrained by the fixed instruction patterns within their training data, and consequently struggle to generalize well even with massive training datasets.

In contrast, agent-based schemes utilize MLLMs to explicitly interpret editing instructions and invoke editing tools to execute the requested modifications. Prior research (Wang et al., 2024; Xue et al., 2025) often relies on curated prompts to guide the agents, but such prompt-driven approaches lack a deep understanding of tools, impeding agents' ability to organize them effectively. Subsequent studies (Guo et al., 2025) attempt to alleviate this burden by constructing predefined workflows as task-level tools for agents to invoke, as shown in Fig. 2 (b). However, this strategy inherently limits the framework's capacity to handle instructions that deviate from the predefined workflows.

To address the challenge of processing flexible real-world instructions for image editing, we propose a novel framework, named Lego-Edit. It employs a fine-tuned MLLM as an agent, termed Builder. The Builder leverages its enhanced reasoning capability to organize a set of specialized pre-trained editing models, called Bricks, enabling precise execution of diverse editing instructions, as shown in Fig. 2 (c). It incorporates two key design innovations:

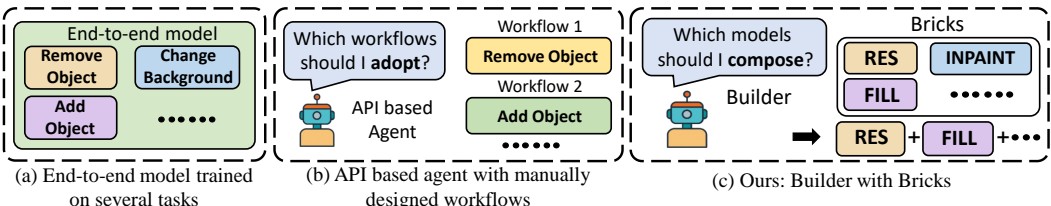

(a) End-to-end model trained on several tasks  (b) API based agent with manually designed workflows  (c) Ours: Builder with Bricks

Figure 2: Comparison of end-to-end approach, API-based agent with curated workflows, and our method.

**Model-Level Tools:** We train and integrate a suite of specialized models as editing tools. This model-level design provides the Builder with high flexibility for composition and enables individual tools to achieve superior performance for their specific functions with less training data.

**Three-Stage Progressive Reinforcement Learning Training Strategy:** We first employ a Supervised Fine-Tuning (SFT) stage, followed by a Reinforcement Learning (RL) stage to train our Builder on specific editing tasks with the ground truth, establishing fundamental reasoning capabilities and tool usage knowledge. Subsequently, we conduct an additional RL stage utilizing abundant unlabeled instructions beyond the specific tasks, where a large-scale critic model provides feedback. This process enhances our Builder's capability to handle flexible instructions.

Benefiting from these designs, our Lego-Edit reliably reasons about and executes flexible editing instructions and can integrate novel tools without additional training. Furthermore, it achieves state-of-the-art results on the GEdit-Bench and ImgBench benchmarks.

In summary, our principal contributions are threefold:

- We propose Lego-Edit, an instruction-based image editing framework that utilizes a reinforcement learning fine-tuned MLLM agent to coordinate model-level editing tools for executing flexible real-world instructions.
- We introduce a three-stage progressive reinforcement learning training strategy that provides feedback using unlabeled data, significantly enhancing the reasoning and tool composition capabilities of MLLM.

- Extensive experiments demonstrate that Lego-Edit achieves SOTA performance on GEdit-Bench and ImgBench. The framework also exhibits strong generalization in processing flexible open-domain instructions and could integrate new tools without retraining.

## 2 RELATED WORKS

**Instruction-based Image Editing:** Instruction-driven image editing, which emerged from Instruct-Pix2Pix (Brooks et al., 2023), provides a natural user interface for image editing. The majority of existing methods rely on diffusion models (Gao et al., 2025; Han et al., 2024; Hertz et al., 2022); while some methods incorporate Multi-modal Large Language Models (MLLMs) to achieve more precise edits like SmartEdit (Huang et al., 2024).

Models trained on massive datasets (e.g., FLUX (Labs, 2024a), HiDream (Cai et al., 2025), Step1X (Liu et al., 2025)) exhibit strong performance across various editing tasks. However, generalizing effectively across flexible editing instructions remains a critical challenge for them. Recently developed unified visual understanding and generation models (e.g., ILLUME++ (Huang et al., 2025a), GPT-4o (Hurst et al., 2024), Bagel (Deng et al., 2025), UniWorld (Lin et al., 2025)), trained on broader datasets encompassing multiple tasks like image captioning and image editing, demonstrate enhanced generalization capabilities yet are still constrained.

In contrast, our proposed approach achieves superior instruction generalization with minimal training data by orchestrating specialized editing tools via a fine-tuned MLLM.

**MLLMs as Agents:** Autonomous agents capable of utilizing tool calls have garnered significant research interest. Many approaches primarily leverage LLMs to invoke tools (Qin et al., 2023; Du et al., 2024; Zheng et al., 2024). With advances in MLLMs, considerable effort has focused on employing or fine-tuning MLLMs for multi-modal agent applications (Huang et al., 2025b).

Some researchers also explore MLLMs as agents for image generation and editing. ComfyAgent (Xue et al., 2025) adapts prompts to call ComfyUI tools via code execution, but this code-based invocation constrains its performance. ComfyMind (Guo et al., 2025) manually defines multiple pipelines for agent-driven tool use, ensuring high success rates but limiting operational flexibility.

Notably, our approach employs reinforcement learning to equip the MLLM with compositional tool-usage knowledge and reasoning ability. Combined with model-level tools, this framework achieves high success rates, superior editing performance, and robust generalization to diverse instructions.

## 3 METHOD

We introduce Lego-Edit, a framework designed for general instruction-based image editing. It uses Builder (an MLLM) to invoke Bricks (model-level tools) for flexibility and employs reinforcement learning (RL) to enhance the Builder's reasoning and tool composition ability. We first outline the overall framework. Then, we describe the tool classification. Finally, we elaborate on the three-stage progressive reinforcement learning (RL) training strategy for the Builder.

### 3.1 OVERALL FRAMEWORK

As illustrated in Fig. 3, our system comprises: 1) the Builder, an MLLM reasoning agent denoted as $f(\theta)$ that generates workflows, where $\theta$ denotes the model parameters; 2) an Executor $D$ that parses and executes workflows; and 3) the Bricks, an external model-level tool library $\mathcal{M} = \{M_1, \ldots, M_N\}$ containing functions encapsulating models or logic processes, where $N$ is the total number of tools.

Given an input pair $(I, q)$ comprising the input image $I$ and editing prompt $q$, the Builder $f(\theta)$, observing the state $s = (I, q)$, generates a reasoning trace, denoted as $Think$, and a JSON-formatted workflow $g$ based on its strategy $\pi_\theta(g \mid s)$. This workflow $g = (V, E)$ is a tool invocation graph. The vertex set $V = \{M_1, ..., M_K\}$ represents the selected tool instances, with each $M_i \in \mathcal{M}$ and $1 \leq K \leq N$; here, $K$ is adaptively determined by the task complexity. The edge set $E \subseteq V \times V$ defines dependencies, where an edge $(M_i, M_j)$ indicates that the input of $M_j$ depends on the output of $M_i$. The Executor $D$ then parses $g$, invokes the tools, and generates the edited image $I' = D(g, I)$.

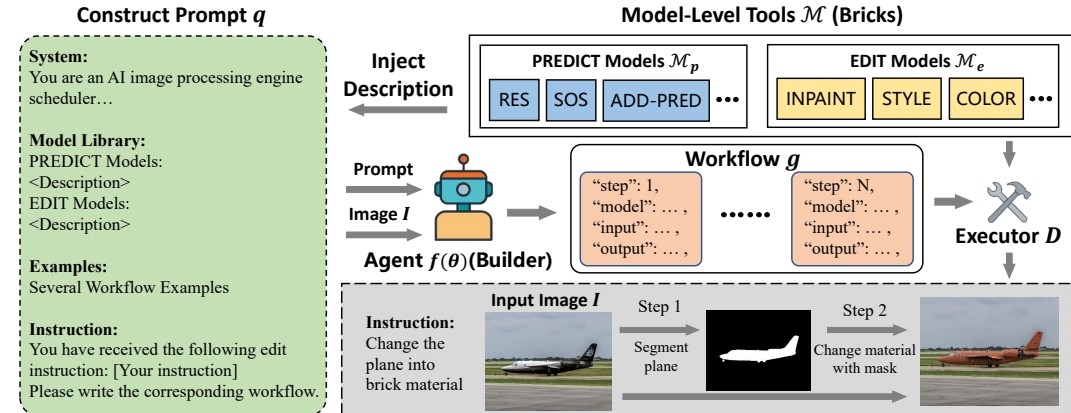

Figure 3: Overall framework of Lego-Edit. Given an instruction and an input image, the Builder generates a tool invocation workflow. The Executor then executes this workflow, calling corresponding tools to generate the edited output image.

## 3.2 MODEL-LEVEL EDITING TOOLS

We constructed a fine-grained library of model-level editing tools $\mathcal{M}$, where each tool represents a single model or function. Tools are categorized into two classes by whether they modify the image:

Predictive Models ($\mathcal{M}_p \subset \mathcal{M}$): Extract/process masks and regions to provide spatial constraints without altering pixels. Included tools are RES (segment specified objects), SOS (subject object segmentation), ADD-PRED (predict addition location), CAP-PRED (image captioning), INVERSE (invert the mask), and one additional tool.

Editing Models ($\mathcal{M}_e \subset \mathcal{M}$): Modify image content. Included tools are FILL (add object with given prompt or reference image), five specialized LoRA adapters trained on FILL: INPAINT (inpainting), POSE (human pose change), ENV (environment alteration), STYLE (style transfer), RCM (material/color change), and three additional tools.

To call a tool, the Builder needs to specify its name, input parameters, and output parameters. Complete details are in the supplementary material.

To prevent task confusion that can arise from joint training in end-to-end models (as in ICEdit (Zhang et al., 2025)), we train independent LoRA adapters for each editing model. And the Builder could precisely control edit scope using masks from $\mathcal{M}_p$, enabling more accurate editing.

## 3.3 THREE-STAGE PROGRESSIVE REINFORCEMENT LEARNING STRATEGY

To train the Builder $f(\theta)$ for effective tool composition, we employ a three-stage progressive RL strategy, gradually increasing task complexity and reducing reliance on ground truth data.

### 3.3.1 REINFORCEMENT LEARNING WITH GRPO

We first introduce the Group Relative Policy Optimization (GRPO) (Shao et al., 2024) algorithm utilized in stages 2 and 3. For a given input $s = (I, q)$, the policy $\pi_\theta$ samples $G$ workflows $\{g_1, ..., g_G\}$. Each workflow $g_j$ receives a reward $r_j$. The relative advantage for each sample within the group is computed as:

$$A_j = \frac{r_j - \text{mean}(\{r_1, \ldots, r_G\})}{\text{std}(\{r_1, \ldots, r_G\})}, \quad j = 1, 2, \ldots, G \tag{1}$$

The policy is updated by maximizing the GRPO objective:

$$\mathcal{J}(\theta) = \mathbb{E}_{s \sim \mathcal{Q}} \mathbb{E}_{\{g_j\}_{j=1}^G \sim \pi_{\theta_{\text{old}}}(\cdot|s)} \left[ \frac{1}{G} \sum_{j=1}^{G} \min \left( r_j^{\text{ratio}} A_j, \text{clip}(r_j^{\text{ratio}}, 1 \pm \epsilon) A_j \right) - \beta D_{\text{KL}}(\pi_\theta \| \pi_{\text{ref}}) \right]$$

$$\tag{2}$$

where $\mathcal{Q}$ denotes the distribution of image-instruction pairs for sampling observation $s$, $\pi_{\theta_{\mathrm{old}}}$ is the old policy before updating, $\pi_{\mathrm{ref}}$ is a fixed reference policy copied from the model's initial state before any training, $r_j^{\mathrm{ratio}} = \pi_\theta(g_j|s)/\pi_{\theta_{\mathrm{old}}}(g_j|s)$, $\epsilon$ controls the clipping range, and $\beta$ weights the KL regularization $D_{\mathrm{KL}}$ towards $\pi_{\mathrm{ref}}$.

### 3.3.2 STAGE 1: SUPERVISED FINE-TUNING (SFT)

We adapt the Builder to image editing domain using SFT on data from several specific tasks. Each sample $(I, q)$ is paired with expert-generated reasoning traces $Think_{\mathrm{GT}}$ and ground truth workflows $g_{\mathrm{GT}}$. The learning target is denoted as the concatenated sequence $l = [Think_{\mathrm{GT}}, g_{\mathrm{GT}}]$. The model is trained to minimize the negative log-likelihood:

$$L_{\mathrm{SFT}} = -\sum_{t=1}^{T} \log p_\theta(l_t \mid I, q, l_{<t}) \tag{3}$$

where $T$ is the total length of $l$, and $p_\theta$ is the model's conditional next-token distribution.

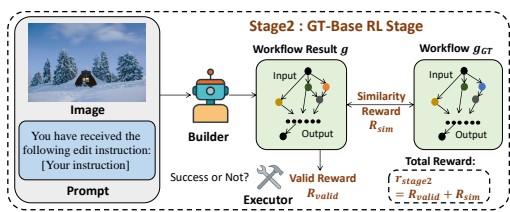

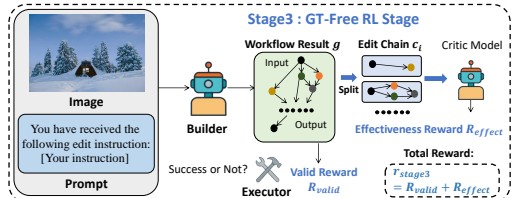

Figure 4: Illustration of the reward design adopted in Stage 2 (GT-based RL training).

Figure 5: Illustration of the reward design adopted in Stage 3 (GT-free RL training).

### 3.3.3 STAGE 2: REFINEMENT WITH GT-BASED REWARDS

Building on SFT, Stage 2 continues training on specific tasks using only $(I, q)$ pairs and $g_{\mathrm{GT}}$ as shown in Fig. 4. The workflow generated by Builder is denoted $g$. We adopt two rewards at this stage:

Valid Reward ($R_{\mathrm{valid}}$): It penalizes non-executable workflows as follows:

$$R_{\mathrm{valid}} = \begin{cases} 0, & \text{if } D \text{ successfully executes } g \\ -1, & \text{otherwise} \end{cases} \tag{4}$$

Similarity Reward ($R_{\mathrm{sim}}$): It measures alignment between $g$ and the expert workflow $g_{\mathrm{GT}}$ using hierarchical graph matching. The depth of nodes in $g$ is calculated based on inverse topological order, which means the depth of the output node is 0. Matched nodes $M$ are found per depth layer using the Hungarian algorithm based on node similarity with a threshold of 0.6. The node similarity $\mathrm{sim}_{\mathrm{node}}$ is calculated by averaging the indicator of whether the same model is used and the proportion of identical parameters. $R_{\mathrm{sim}}$ combines node coverage and average matched node similarity:

$$R_{\mathrm{sim}} = 0.5 \cdot \frac{|M|}{\max(|V_g|, |V_{g_{\mathrm{GT}}}|)} + 0.5 \cdot \frac{1}{|M|} \sum_{(i,j) \in M} \mathrm{sim}_{\mathrm{node}}(i, j) \tag{5}$$

where $V_g$ and $V_{g_{\mathrm{GT}}}$ denote the node sets of the generated workflow $g$ and the ground truth workflow $g_{\mathrm{GT}}$ respectively.

The total reward is: $r_{\mathrm{stage2}} = R_{\mathrm{valid}} + R_{\mathrm{sim}}$.

### 3.3.4 STAGE 3: GENERALIZATION WITH GT-FREE CRITIC REWARDS

Stage 3 targets generalization to open-domain instructions using only $(I, q)$ pairs. We employ the same valid reward $R_{\mathrm{valid}}$ as Eq. 4 and another effectiveness reward to provide feedback without a ground truth workflow as shown in Fig. 5:

Effectiveness Reward ($R_{\mathrm{effect}}$): It uses an MLLM critic model $C$ to assess semantic alignment between the workflow's effect and instruction $q$. Workflows are decomposed into several editing chains

$c_i$, each of which contains only one editing model in $\mathcal{M}_e$ to perform actual editing. The critic $C$ would abstract each chain's effect into a meta-edit description $m_i$. Then it evaluates the description set $\{m_i\}$ against $q$, which is formulated as:

$$(N_{\text{remove}}, N_{\text{add}}, A) = C(I, q, \{m_i\}) \tag{6}$$

Specifically, critic $C$ must determine whether to remove existing editing chains or add new editing chains to better achieve the instruction. $N_{\text{remove}}$ denotes the number of chains to remove, $N_{\text{add}}$ denotes the number of chains to add and $A$ denotes the content of added new chains. $R_{\text{effect}}$ applies a penalty defined as follows:

$$R_{\text{effect}} = 1 - 0.5 \cdot (N_{\text{add}} + N_{\text{remove}}) \tag{7}$$

The total reward is: $r_{\text{stage3}} = R_{\text{valid}} + R_{\text{effect}}$.

## 4 EXPERIMENTS

We first demonstrate the model's zero-shot capability, followed by extensive experiments validating the superiority of our framework on image editing benchmarks. Then we analyze the performance improvements and other advantages introduced by model-level tools, and finally demonstrate the Builder's improved generalization and performance enabled by reinforcement learning.

### 4.1 IMPLEMENTATION DETAILS

**Builder:** Our Builder is based on MiMo-VL-7B (Team et al., 2025) and undergoes full-parameter fine-tuning in bf16 using a progressive three-stage curriculum: (1) 500 image-text pairs (instructions, thoughts and workflows), (2) 10K pairs (instructions and workflows), and (3) 50K pairs (instructions), all sourced from OmniEdit (Wei et al., 2024). The generation of thoughts and workflows, along with the critic model used in stage 3 training, are both based on Qwen2.5-VL-72B (Bai et al., 2025). Each stage trains for 1 epoch with AdamW (lr = 1e−5, $\beta = (0.9, 0.999)$, weight decay = 0.01), without warmup or decay (batch size 8, image size $448 \times 448$). Data construction details are provided in the supplementary material.

**Predictive Tools:** RES utilizes EVF-SAM (Zhang et al., 2024), trained from scratch on 200K MS COCO (Lin et al., 2014) samples and 8K HierText (Long et al., 2022) samples (lr = 1e−4, batch size 64, resolution $512 \times 512$, 10K iterations) with BCE and Dice loss. SOS employs U2Net (Qin et al., 2020), trained on 5K DIS (Qin et al., 2022) with identical loss and optimization settings (batch size 24, 100 epochs, BCE loss only). Both models initialize without pretrained weights. ADD-PRED and CAP-PRED are built upon our Builder. CAP-PRED directly leverages the Builder's native captioning capability. ADD-PRED is fine-tuned with LoRA adapters (rank=32) on 50K OmniEdit (Wei et al., 2024) samples for addition region prediction, formulated as bounding box generation between the source and target images, training uses 1 epoch and a learning rate of 1e−4.

**Editing Tools:** We adopt ICEdit's (Zhang et al., 2025) framework (LoRA fine-tuning at rank=32) but implement five specialized adapters for individual tasks, rather than the multi-task adapter. Each adapter is trained on 10K task-specific samples curated from OmniEdit (Wei et al., 2024) and MagicBrush (Zhang et al., 2023) via VIEScore (Ku et al., 2023) assessment, using a global batch size of 8 for 10K steps at 768×768 resolution. We train these tools on two backbones, FLUX-1 Fill (Labs, 2024b) and Qwen-Image-Edit Wu et al. (2025a). In addition, we incorporate the open-sourced LoRA of FLUX-Text (Lan et al., 2025) into our toolbox.

All the experiments utilized 8×NVIDIA H20 GPUs for training. We adopt DeepSpeed ZeRO-3 (Rajbhandari et al., 2020) to accelerate training.

### 4.2 EVALUATION SETTINGS

To ensure authoritative evaluation, we benchmark our method on two widely adopted datasets: GEdit-Bench (606 samples) (Liu et al., 2025) and ImgEdit-Bench (811 samples) (Ye et al., 2025), known for complex editing instructions and high-quality imagery. Following standard protocols, we employ VIEScore (Ku et al., 2023) executed by GPT-4o (Hurst et al., 2024) as our primary metric. To ensure fairness and reproducibility, we fix the random seed to 0 and perform single-shot inference for all evaluations. We evaluate both the FLUX-Tools and Qwen-Tools variants of our system, while all ablation studies are performed using the FLUX-Tools variant.

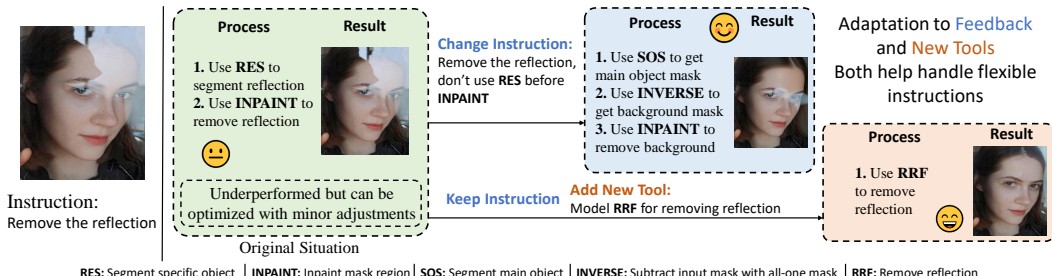

Figure 6: Comparison with other methods on complex edits (top) and our Tool composition workflow (bottom).

### 4.3 ZERO-SHOT CAPABILITY OF THE BUILDER

**Zero-Shot Complex Edits with Flexible Tool Composition:** Fig. 6 presents a visual comparison of editing results on flexible instructions, alongside the Builder's tool composition process. For the "swap" instruction, although the Builder was not explicitly trained on this task, it effectively decomposes the instruction into atomic operations by first removing object A using RES and INPAINT, then inserting object B via ADD-PRED and FILL. This example shows its ability to compose specialized tools for flexible editing instructions, which enables complex edits beyond the reach of end-to-end or curated-pipeline models.

Figure 7: Example of Zero-Shot adaptation via feedback and Tool Insertion in reflection removal.

**Zero-Shot Adaptation to Feedback and New Tools:** Fig. 7 demonstrates the Builder's adaptability to user feedback and new tools without retraining. For the reflection removal task, the Builder's initial workflow (RES and INPAINT) failed because RES could not segment reflections effectively. Users can provide feedback, such as "don't use RES before INPAINT" to prevent it. The Builder will revise its workflow: it uses SOS for foreground segmentation, INVERSE to acquire the background, and then INPAINT to remove part of the reflection. Additionally, users can introduce a dedicated reflection-removal tool (RRF), which the Builder readily adopts to solve the task effectively. This illustrates the system's adaptability to extend capabilities by integrating new tools or incorporating user feedback, all without modifying the Builder.

### 4.4 COMPARISON WITH STATE-OF-THE-ART

**GEdit-Bench:** As shown in Tab. 1, the FLUX-Tools variant achieves competitive overall performance (G_O = 6.88), while the Qwen-Tools variant achieves the highest performance across all three metrics, outperforming recent SOTA methods Qwen-Image-Edit and SeedEdit-4.0. Fig. 8 further illustrates the FLUX-Tools variant's strength in fine-grained tasks such as color change and material replacement. This precision stems from the Builder's ability to compose RES, enabling localized edits while preserving non-target regions (Sec. 4.6). Compared to traditional API agents like ComfyMind that rely on pre-scripted pipelines, Lego-Edit gains notable advantages through its capable Builder and flexible orchestration of specialized Tools.

**Generalization Beyond Training Tasks:** In the first two training stages, the Builder is trained on five tasks (Background Change, Color Alteration, Subject Addition, Subject Removal, and Subject Replacement). During evaluation, it generalizes effectively to six additional tasks. This generalization is enabled by the Builder's flexible composition of Tools, which allows it to tackle untrained tasks. As shown in Fig. 8, the FLUX-Tools variant achieves strong results on most untrained tasks, demonstrating robust generalization beyond its training scope.

Table 1: Quantitative evaluation on GEdit-Bench-EN. All metrics are reported as higher-is-better (↑).

| Methods | G_SC↑ | G_PQ↑ | G_O↑ |
|---|---|---|---|
| Instruct-P2P | 3.58 | 5.49 | 3.68 |
| MagicBrush | 4.68 | 5.66 | 4.52 |
| AnyEdit | 3.18 | 5.82 | 3.21 |
| OmniGen | 5.96 | 5.89 | 5.06 |
| Step1X-Edit | 7.13 | 7.00 | 6.44 |
| BAGEL | 7.36 | 6.83 | 6.52 |
| UniWorld-V1 | 4.93 | 7.43 | 4.85 |
| ComfyMind | 2.67 | 5.93 | 2.61 |
| OmniGen2 | - | - | 6.42 |
| FLUX.1 Kontext | - | - | 6.26 |
| Qwen-Image-Edit-2509 | 8.15 | 7.86 | 7.54 |
| SeedEdit-4.0 | 8.17 | 7.66 | 7.44 |
| **Ours-FLUX-Tools** | 6.45 | 7.45 | 6.88 |
| **Ours-Qwen-Tools** | **8.42** | **7.90** | **7.84** |

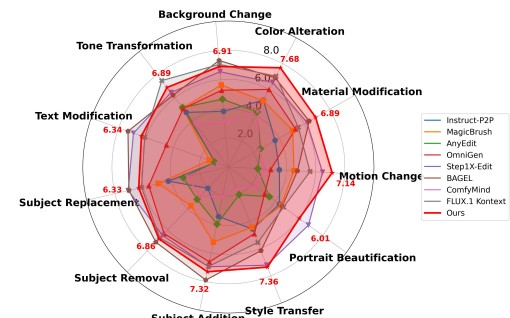

Figure 8: VIEScore of Each Sub-task in GEdit-Bench-En of FLUX-Tools variant. All the results are evaluated by GPT-4o.

Table 2: Quantitative evaluation on ImgEdit-Bench. All metrics are reported as higher-is-better (↑).

| Methods | Add | Adjust | Extract | Replace | Remove | Style | Action | Hybrid | Background | Overall↑ |
|---|---|---|---|---|---|---|---|---|---|---|
| MagicBrush (Zhang et al., 2023) | 2.72 | 1.47 | 1.31 | 1.89 | 1.57 | 2.49 | 1.39 | 1.80 | 2.03 | 1.85 |
| Instruct-P2P (Brooks et al., 2023) | 2.29 | 1.79 | 1.33 | 1.93 | 1.49 | 3.54 | 1.51 | 1.48 | 1.67 | 1.89 |
| AnyEdit (Yu et al., 2025) | 3.12 | 2.66 | 1.82 | 2.71 | 2.34 | 3.27 | 3.31 | 2.07 | 2.37 | 2.63 |
| UltraEdit (Zhao et al., 2024) | 3.63 | 3.01 | 2.02 | 3.13 | 1.71 | 3.69 | 3.57 | 2.33 | 3.31 | 2.93 |
| Step1X-Edit (Liu et al., 2025) | 3.90 | 3.13 | 1.87 | 3.45 | 2.61 | 4.44 | 3.43 | 2.52 | 3.19 | 3.17 |
| BAGEL (Deng et al., 2025) | 3.55 | 3.30 | 1.56 | 3.38 | 2.44 | 4.24 | 4.29 | 2.55 | 3.22 | 3.17 |
| UniWorld-V1 (Lin et al., 2025) | 3.86 | 3.70 | 2.23 | 3.49 | 3.54 | 4.22 | 3.44 | 3.13 | 2.76 | 3.37 |
| ComfyMind (Guo et al., 2025) | 1.45 | 3.14 | 2.21 | 3.43 | 2.81 | 2.66 | 2.74 | 0.57 | 2.26 | 2.63 |
| OmniGen2 (Wu et al., 2025b) | 3.57 | 3.06 | 1.77 | 3.74 | 3.20 | 4.81 | 4.68 | 2.52 | 3.57 | 3.44 |
| FLUX.1 Kontext (Batifol et al., 2025) | 3.76 | 3.45 | 2.15 | 3.98 | 2.94 | 3.78 | 4.38 | 2.96 | 4.26 | 3.52 |
| Qwen-Image-Edit-2509 (Wu et al., 2025a) | **4.32** | 4.36 | 4.04 | **4.64** | 4.52 | 4.84 | 4.71 | 3.39 | 4.37 | 4.35 |
| SeedEdit-4.0 (Seedream et al., 2025) | 4.17 | 4.35 | 3.87 | 4.43 | 4.66 | 4.70 | 4.68 | 3.51 | **4.49** | 4.32 |
| **Ours-FLUX-Tools** | 4.03 | 3.84 | 2.47 | 3.41 | 3.42 | 4.48 | 4.04 | 3.20 | 3.41 | 3.59 |
| **Ours-Qwen-Tools** | 4.19 | **4.41** | **4.17** | 4.58 | **4.70** | **4.86** | **4.74** | **3.60** | 4.22 | **4.39** |

**ImgEdit-Bench:** Our FLUX-Tools variant maintains top performance on ImgEdit-Bench, while the Qwen-Tools variant achieves the highest overall score (4.39) among all compared methods including recent SOTA methods Qwen-Image-Edit and SeedEdit-4.0. The detailed results are shown in Tab. 2. Crucially, our Qwen-Tools variant dominates the most challenging Hybrid Editing sub-task (3.60). This success validates our proposition that the Builder can parse composite instructions into atomic sub-tasks and dynamically generate workflows to coordinate specialized Tools.

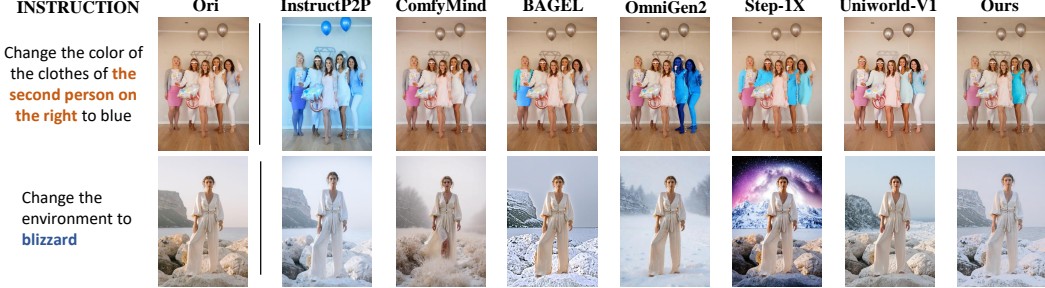

Figure 9: Qualitative results compared to other methods.

**Qualitative Results:** As illustrated in Fig. 9, our method outperforms other approaches in both edit accuracy and visual realism. Edits are well-aligned with the intended regions and maintain high perceptual quality. More comparative results are provided in the supplementary material.

## 4.5 Ablation Study on Tools

**Necessity of Task-Specialized Tools.** To validate Lego-Edit's design, we compare its task-specialized architecture with a unified alternative using identical settings. Three separate LoRA adapters are trained on 10K samples per task (Color Alteration, Style Transfer, Tone Transfer), while the unified model uses a combined 30K dataset. As shown in Tab. 3, specialized models outperform the unified one (e.g., 6.83 vs. 5.94 in color alteration). Increasing LoRA rank in the unified model brings no gain. Qualitative results (provided in supplementary material) reveal frequent task confusion in the unified setup, highlighting the importance of specialization for editing fidelity.

Table 3: Comparison between Multi-task LoRA and Single-task LoRA on GEdit-Bench-EN, we report the G_O metric.

| Model | Rank | Color Alteration | Style Transfer | Tone Transfer |
|---|---|---|---|---|
| Multi-task | 32 | 5.94 | 5.47 | 5.56 |
| Multi-task | 64 | 5.42 | 5.43 | 5.57 |
| Single-task | 32 | **6.83** | **6.75** | **6.63** |

Table 4: Comparison between separately adopted and composed adopted tools on GEdit-Bench-EN.

| Task | Compose Pattern | G_O |
|---|---|---|
| Color Alteration | RCM | 6.83 |
| | RCM + RES | **7.68** |
| Material Modification | RCM | 6.08 |
| | RCM + RES | **6.89** |
| Style Transfer | STYLE | 6.75 |
| | STYLE + CAP-PRED | **7.36** |

## 4.6 Ablation Studies on Builder

Table 5: Performance is evaluated by execution success rate, indicating error-free syntax, and VIEScore, assessing editing quality. Success rate comprises Simple %Pass on original instructions and Complex %Pass on three GPT-4o-generated variants per instruction.

| Agent Model | Simple %Pass | Complex %Pass | G_SC↑ | G_PQ↑ | G_O↑ |
|---|---|---|---|---|---|
| GPT-4o | 75.2 | 52.1 | 4.28 | 6.53 | 5.10 |
| MiMo-VL-7B-Base | 56.8 | 44.9 | 3.11 | 4.66 | 3.59 |
| MiMo-SFT | 75.7 | 53.8 | 4.39 | 7.01 | 5.22 |
| MiMo-RL w/ GT | **100** | 81.9 | 5.49 | 7.35 | 6.16 |
| MiMo-RL w/o GT | **100** | **99.1** | **6.45** | **7.45** | **6.88** |
| Qwen3-VL-2B-Base | 49.6 | 39.8 | 2.74 | 6.01 | 3.12 |
| Qwen3-SFT | 68.3 | 47.1 | 4.17 | 6.58 | 4.99 |
| Qwen3-RL w/ GT | 94.8 | 73.7 | 5.33 | 7.32 | 6.09 |
| Qwen3-RL w/o GT | **100** | 98.5 | 6.40 | 7.41 | 6.85 |

**Effectiveness of Reinforcement Learning Training.** Ablations on GEdit-Bench in Tab. 5 show the effectiveness of our progressive RL training. Starting from Builder-SFT, which outperforms basemodel MiMo-VL-7B and powerful MLLM GPT-4o on simple and complex success rates and VIEScores, subsequent RL training with ground truth (Builder-RL w/ GT) achieves 100% simple and 81.9% complex success rate with better VIEScores. Final GT-free RL training (Builder-RL w/o GT) maintains 100% simple success, boosts complex success to 99.1%, and achieves the highest VIEScores. Furthermore, we adopt Qwen3-VL-2B as the Builder base model and observe that, although its initial capability is weaker than MiMo, after the same three-stage training pipeline its performance becomes comparable to MiMo, highlighting the strong effectiveness and robustness of our training methodology.

**Latency Discussion.** We evaluate our system on GEdit-Bench using a single H20 GPU. The MiMo-based Builder takes 3.5 seconds, and the slowest Tool takes 2.7 seconds, resulting in a total pipeline latency of approximately 6.9 seconds. In comparison, the Qwen3-VL-based Builder takes 1.9 seconds, with the slowest Tool still at 2.7 seconds, yielding a total latency of approximately 5.2 seconds. for the end-to-end method, Bagel takes 25 seconds, Step1X-Edit takes 21.3 seconds, Omnigen2 takes 22.6 seconds and FLUX-Kontext takes 7.4 seconds, all under identical settings.

**Effectiveness of Tool Composition.** To evaluate the impact of explicit Tool composition in Lego-Edit, we examine its performance across GEdit-Bench sub-tasks. As shown in Tab. 4, integrating RES segmentation masks with RCM increases G_O for color alteration and material modification, highlighting the benefits of RES's spatial control. Similarly, incorporating CAP-PRED text descriptions into STYLE boosts G_O in style transfer, as textual cues enforce semantic alignment between stylized outputs and the source image.

## 5 LIMITATIONS

For most suboptimal outcomes, the primary limiting factor was the capability of the toolset itself. We summarize two major sources of failure: insufficient tool performance and toolset incompleteness.

### 5.1 INSUFFICIENT TOOL PERFORMANCE

Insufficient tool performance refers to cases where the Builder reasonably invokes tools, but the failure of a specific tool leads to an incorrect outcome. As illustrated in Fig. 10, our instruction is "set the clock to 12:10". Neither our framework, Qwen-Image-Edit, nor SeedEdit-4.0 could accomplish this task. For our framework, the Builder gives a clear workflow: it first calls the RES model to segment and remove the hour hand, then predicts the position of the hand at 12:10 and inserts it. The left part of ours in Fig. 10 shows that the process fails because the RES model incorrectly extracts the entire clock face. We fine-tune the RES model with a few hour-hand examples to improve hand identification ability. As shown in the right part of ours in Fig. 10, once the hour hand is correctly segmented and removed, the subsequent steps succeed, ultimately fulfilling the instruction.

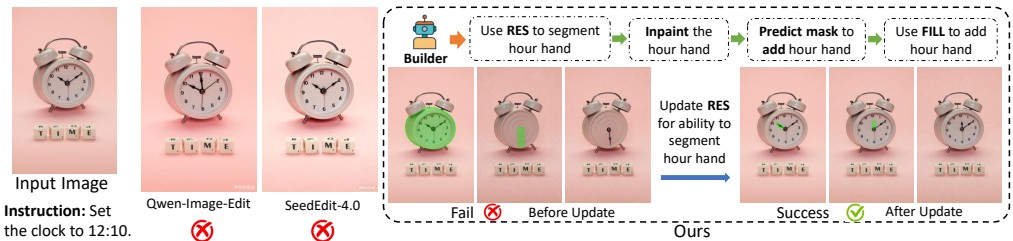

Figure 10: Example of editing failures caused by insufficient tool performance.

### 5.2 TOOLSET INCOMPLETENESS

Toolset incompleteness occurs when available tools cannot fully satisfy a user instruction, so the Builder produces the closest possible workflow, but results remain limited. As illustrated in Fig. 11, we instructed the model to extend the image to achieve a balanced composition. Both Qwen-Image-Edit and SeedEdit-4.0 failed to follow the instruction, while our framework produced a slightly better result. As shown in the left part of ours in Fig. 11, given the current toolset, the Builder correctly inferred the downward direction but overextended the image by a factor of 1.5, producing more padding than necessary. To address this, we trained a LoRA based on Builder on 2k examples to predict precise extension ratios in four directions. The Builder invokes this tool to accurately estimate the required extension ratio and ultimately satisfies the instruction.

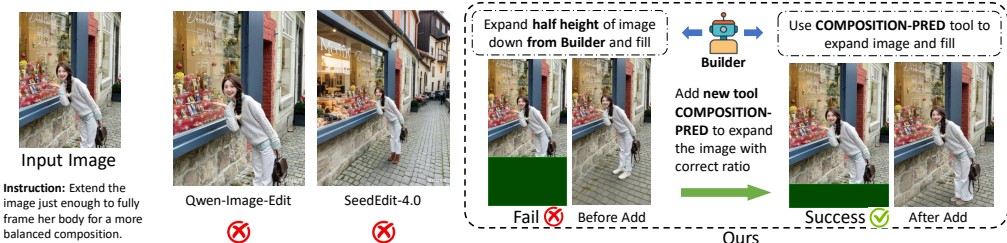

Figure 11: Example of editing failures caused by toolset incompleteness.

## 6 CONCLUSION

In this paper, we introduce Lego-Edit, a novel agent-based framework for instruction-based image editing. It employs an RL-finetuned MLLM (Builder) to orchestrate model-level editing tools (Bricks), enabled by two key innovations: fine-grained tools allowing flexible composition and precise control, and a three-stage progressive RL training strategy enhancing reasoning and tool organization abilities via GT-free feedback. Extensive experiments demonstrate Lego-Edit's state-of-the-art performance on GEdit-Bench and ImgBench, showcasing superior accuracy and generalization in handling flexible requests and integrating new tools without retraining. Future work will expand the tool set to fully leverage the extensibility of Builder.

REPRODUCIBILITY STATEMENT

We fully recognize the challenges of reproducibility in current AI research and have taken concrete actions to address them. First, we have provided an anonymized code repository in the supplementary materials. Second, we devote a substantial section to implementation details, thoroughly documenting all training parameters and methodologies for the models. Finally, to ensure a comprehensive understanding and control over every component of our system, we have retrained the majority of our tools, even though they are based on open-source methods. This process can be regarded as both a validation and refinement of existing approaches, and the resulting models may serve as new baselines for the community.

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

# A    DETAILS OF MODEL-LEVEL TOOLS

We provide details of all model-level tools in Tab. 6, including their names, functions, inputs, and outputs.

Table 6: Details of Model-Level Tools: Names, functions, inputs, and outputs.

| Tool Type | Tool Name | Function | Input | Output |
|---|---|---|---|---|
| **Predictive Models** | RES | Segmentation of object specified by prompt | Image[image], Str[prompt] | Mask[mask], Image[image] |
| | SOS | Segmentation of main objects in image | Image[image] | Mask[mask], Image[image] |
| | ADD-PRED | Given a prompt and a mask. If mask=null, predict the most appropriate position to add the target represented by this prompt to the image, If mask!=null, this position must be within the given mask | Image[image], Str[prompt], Mask[mask] | Mask[mask] |
| | CAP-PRED | Describe the image in English, this description is applied to the FLUX model, so that the generated image is inspired by the original image. If it is an image expansion task, the input ratio needs to be given so that the output image and the output mask is output and applied to the FLUX model. Otherwise, the input ratio=null and the output mask=null, the output image=null | Image[image], Float[left_ratio], Float[right_ratio], Float[top_ratio], Float[bottom_ratio] | Str[caption], Image[image], Mask[mask] |
| | INVERSE | Subtract mask2 from mask1 or subtract image2 from image1, if mask1 is null it means use a mask with all pixels to be 1 minus the input mask2 | Mask[mask1], Mask[mask2], Image[image1], Image[image2] | Mask[mask], Image[image] |
| | BBOX | Given a mask, output the bounding box mask of it | Mask[mask] | Mask[mask] |
| **Editing Models** | FILL | Generated in the mask area according to the specified prompt, don't use to replace the color or material of an object | Image[image], Mask[mask], Str[prompt], Image[preimage] | Image[image] |
| | FASTINPAINT | For quick inpaint and the score of the inpaint effect will be output | Image[image], Mask[mask] | Image[image], Float[score] |
| | INPAINT | Fill background in mask area, generate reference the input preimage and the score | Image[image], Mask[mask], Image[preimage], Float[score] | Image[image] |
| | RCM | Replace the color or material of an object | Image[image], Mask[mask], Str[prompt] | Image[image] |
| | STYLE | Convert the style of the input image or a specific object in the image, you must give an input style, such as 'anime style' | Image[image], Mask[mask], Str[prompt], Str[style] | Image[image] |
| | ENV | Replace the environment of an object, like the weather, the climate, or the times of day | Image[image], Str[prompt] | Image[image] |
| | POSE | Change the object's posture, expression, etc. | Image[image], Str[prompt] | Image[image] |
| | COMPOSE | Compose two input masks or images, if both have values at same pixels, the second input will cover the first | Mask[mask1], Mask[mask2], Image[image1], Image[image2] | Mask[mask], Image[image] |
| | RESIZE | Resize the width and height of the valid part of input mask or image to the given ratio times original width and height | Mask[mask], Image[image], Float[ratio] | Mask[mask], Image[image] |
| | TEXT | Add given text to the image with given mask | Image[image], Mask[mask], Str[text] | Image[image] |

# B    ADDITIONAL COMPARISONS WITH STATE-OF-THE-ART METHODS

We conduct a series of additional evaluations to comprehensively compare our system with state-of-the-art methods. In all subsequent experiments, results are presented under the FLUX-Tools setting unless otherwise noted as Qwen-Tools.

## B.1    HUMAN EVALUATION ON STATE-OF-THE-ART METHODS

To compare our approach with other methods, we conducted a ranking-based human evaluation. The evaluation set consists of 500 test cases, constructed from a combination of volunteer-provided images and internally collected images. This ensures that the set covers not only flexible and challenging editing tasks but also real-world editing scenarios. Each test case comprises an input image and an associated editing instruction. Five volunteers, blind to the method identities, were presented

with the editing results of all six methods for each test case and asked to rank them from best (1) to worst (6) according to how well the instruction was satisfied and whether the non-edited regions were consistent. To prevent position bias, the display order of the results was randomized across cases. Each method thereby received a rank for every test case, and the final score was computed as the average rank across all 500 cases. Since the average rank ranges between 1 (best) and 6 (worst), lower values indicate better performance. Table 7 reports the results. Notably, our method obtains the lowest average rank across 500 test cases in the human evaluation, providing strong evidence of its superiority over state-of-the-art methods.

Table 7: The average rank of each method in human evaluation.

| Method | Step1X-Edit | BAGEL | UniWorld-V1 | OmniGen2 | FLUX.1 Kontext | Ours |
|---|---|---|---|---|---|---|
| Average Rank↓ | 3.88 | 3.07 | 4.72 | 3.75 | 2.87 | **2.72** |

### B.2 EVALUATION ON CONSISTENCY OF NON-EDITED REGIONS

To further evaluate the consistency of non-edited regions before and after editing between our method and comparative approaches, we adopted the same testing protocol as FLUX-Kontext (Batifol et al., 2025), using facial similarity before and after editing as the metric to assess consistency and adopting AuraFace (Deng et al., 2019) to measure it. Face preservation, being an implicit requirement in image editing, serves as an appropriate measure for such consistency evaluation. As shown in Tab. 8, which presents facial similarity measurements for all methods tested exclusively on GEdit-Bench tasks that do not alter facial attributes, our method (both FLUX-Tools variant and Qwen-Tools variant) achieves the best face preservation performance among all compared approaches. This result also indicates that our method is more effective in maintaining consistency in non-edited regions before and after editing.

Table 8: Facial similarity measurements for all compare methods on sub-tasks of GEdit-Bench-EN. All similarities are between 0 to 1 and higher is better.

| Methods | Background Change | Color Alter | Subject Add | Subject Remove | Overall↑ |
|---|---|---|---|---|---|
| Step1X-Edit | 0.65 | 0.65 | 0.62 | 0.71 | 0.66 |
| BAGEL | 0.74 | 0.71 | 0.73 | 0.65 | 0.71 |
| UniWorld-V1 | 0.55 | 0.61 | 0.65 | 0.59 | 0.60 |
| OmniGen2 | 0.57 | 0.72 | 0.66 | 0.65 | 0.65 |
| FLUX.1 Kontext | 0.85 | 0.79 | 0.71 | 0.74 | 0.77 |
| Qwen-Image-Edit-2509 | 0.82 | 0.84 | 0.85 | 0.79 | 0.83 |
| SeedEdit-4.0 | 0.77 | 0.83 | 0.81 | 0.87 | 0.82 |
| **Ours-FLUX-Tools** | **0.90** | **0.85** | **0.88** | **0.87** | **0.88** |
| **Ours-Qwen-Tools** | **0.92** | **0.85** | **0.90** | **0.88** | **0.89** |

### B.3 MORE QUALITATIVE EXAMPLES

We present extended qualitative comparisons with state-of-the-art methods in Fig. 12. And Fig. 13 showcases results achieved with more complex and flexible instructions. Further examples demonstrating performance across different aspect ratios are shown in Fig. 14.

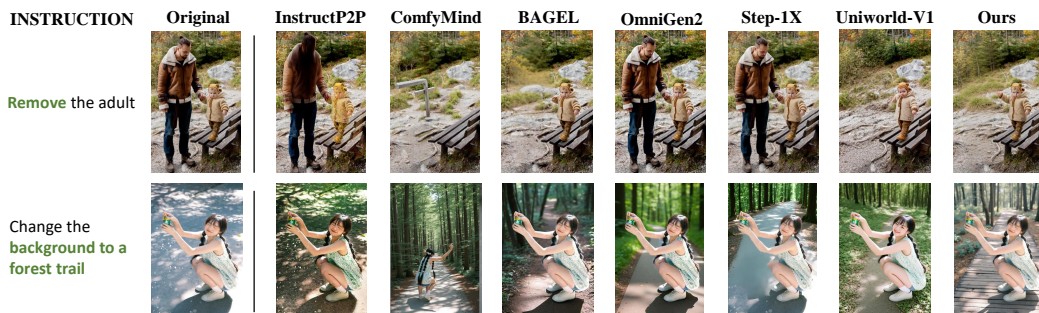

Figure 12: Qualitative comparison with state-of-the-art methods.

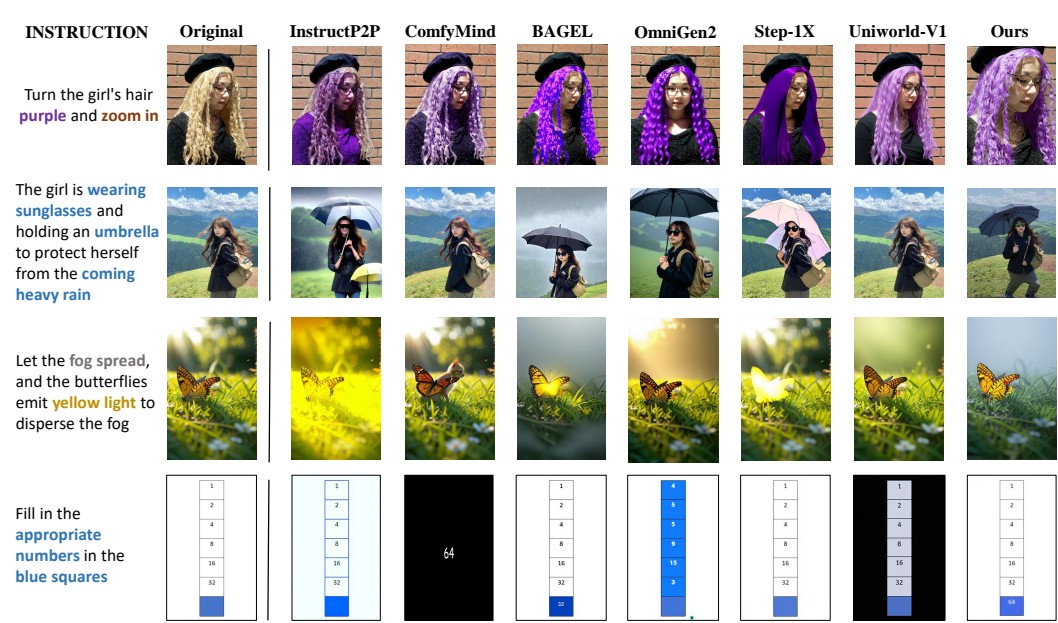

Figure 13: Qualitative results of handling complex and flexible editing instructions.

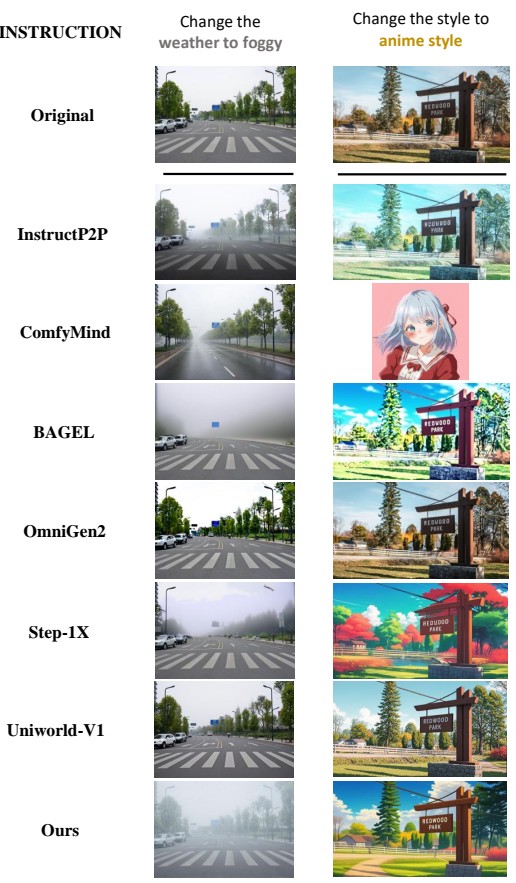

Figure 14: Qualitative results for image editing at varying aspect ratios.

## C DEMONSTRATIONS OF RESPONSE TO FLEXIBLE INSTRUCTION

To more clearly demonstrate Builder's capability in handling flexible instructions through reasoning and tool composition, we provide the editing results along with the complete chain of thought on two test cases with flexible instructions. As illustrated in Fig. 15, Builder is tasked with depicting how the objects in the image would appear after a certain period of time. It is able to correctly reason about the changes the objects would undergo over time and accordingly invoke the appropriate tools. For the banana, which would rot or decay, Builder utilizes the RCM tool to alter its color and texture, simulating the appearance of rot. For the sapling, Builder considers that it would grow into a tree over time and thus employs an additive approach, referencing the original sapling to add a tree to the scene. This allows the editing task to be accurately completed. For the same prediction task, Builder demonstrates precise reasoning and flexibly determines the appropriate tools to invoke based on the practical context, highlighting its capability to handle flexible instructions.

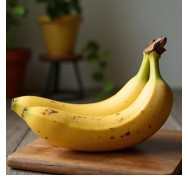

**Input Image**

**INSTRUCTION:**
Draw what it will look like after being kept in a daily environment for a year.

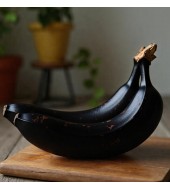

**Output Image**

**CoT of Builder**

Okay, let's tackle this problem. The user wants to know what the bananas will look like after being kept in a daily environment for a year. First, I need to figure out how to approach this using the available tools.

The bananas in the image are fresh, with a yellow color and some brown spots. Over time, they'll likely ripen further, maybe even overripe. **The key changes would be the color darkening, more brown spots, and possibly some softening or even mold if left too long.** But since the instruction is to keep them in a daily environment, maybe they're still being used, so maybe just more spots and darker color.

First, I need to segment the bananas. **The RES model can do that with a prompt like "the bananas." That gives me a mask of the bananas.** Then, I need to apply the changes over a year. **The FLUX-RCM model is for changing color or material. The prompt here would be "darken the color of the bananas and add more brown spots." That should handle the color change and the spots.**

Wait, but FLUX-RCM requires a mask. So after getting the mask from RES, I can use it with FLUX-RCM. The input image is the original, the mask is from RES, and the prompt is the desired change. That should do it. Let me check the constraints again. FLUX-RCM needs the image, mask, and prompt. The prompt should be specific, like "darken the color...". That fits. So the pipeline is RES to get the mask, then FLUX-RCM with the mask and the prompt. That should result in the bananas looking more aged, with darker color and more spots. That makes sense for a year's worth of aging.

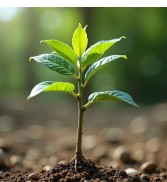

**Input Image**

**INSTRUCTION:**
Draw what it will look like ten years later.

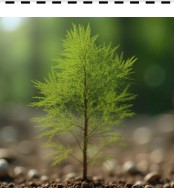

**Output Image**

**CoT of Builder**

Okay, let's tackle this problem. The user wants to know what the plant will look like ten years later. First, I need to figure out which tools to use. The available models include segmentation, prediction, and editing tools.

The plant is the main subject here. To start, I should segment the plant to get a mask. The RES model is good for object segmentation based on a prompt. Using "the plant" as the prompt should work. Once I have the mask, the next step is to predict how the plant will grow in ten years. The ADD-PRED model can help determine the appropriate position and size for the future plant. Since the plant is likely to grow larger and more complex, I need to adjust the mask accordingly.

After getting the predicted mask, I need to generate the new plant. The FLUX-FILL model is suitable here because it can generate content within a mask based on a prompt. **The prompt should describe the mature plant's features, like a thick stem, broad leaves, and a well-developed root system.** Using the original segmented image as a preimage might help maintain consistency in the style and details.

I should check if any other models are needed. The CMI-PRED is for image expansion, but since the plant is within the existing frame, expansion isn't necessary. The FLUX-STYLE isn't needed here because the style is natural, not requiring a style transfer.

**Putting it all together: first, segment the plant with RES, then use ADD-PRED to predict the future growth, and finally use FLUX-FILL to generate the mature plant.** This pipeline should effectively model the plant's development over ten years.

Figure 15: Qualitative results and chains of thought of handling flexible editing instructions.

## D    EXTENSION FOR MASK INPUT

We have observed that in practical editing applications, the target regions of certain edits are often difficult to describe directly in language. Using masks to select the editing region can significantly simplify this process while improving editing accuracy. Therefore, we extended our Builder to support mask input for defining the editing area. Specifically, we made two modifications to the prompt mentioned in Sec. I which is provided to the Builder. First, we incorporated "init[mask]" into the input parameters to enable the Builder to recognize that masks can be provided as part of the input. Second, we added examples of editing workflows that utilize the input "init[mask]" to the Builder's demonstration set, helping it learn how to employ masks appropriately. During actual inference, if the user provides a mask, we prepend a fixed preset instruction to the editing command: "The user has provided a mask and expects to" thereby informing the Builder of the presence of the mask.

Fig. 17 presents the subjective outcomes of mask-based processing across multiple tasks, confirming that the Builder correctly understands and processes the mask inputs. Notably, the method performs effectively in cases where edit localization is particularly challenging. For instance, with the instruction "Remove it" and input mask, the system successfully removes one specific car among multiple vehicles, and for instruction "Add a hat on his head" it accurately places a hat on the intended person within a crowded scene. By incorporating mask input, both challenging scenarios are handled appropriately.

## E    MORE CASES FOR TEXT EDIT

Text modification is a common image editing requirement in practical applications. Therefore, we have extended our framework to support text editing capabilities. Specifically, we integrated FLUX-Text (Lan et al., 2025) into our framework, which is a model trained with LoRA on FLUX for text editing. A new tool function was developed for Builder to add text as required and invoke FLUX-Text for text refinement. With only minimal code modifications, this open-source tool was seamlessly integrated into our framework. Fig. 16 presents some subjective examples of text editing performed after integration, all of which successfully and effectively completed the editing instructions. This seamless integration further demonstrates that our framework can continuously expand the boundaries of its editing capabilities by incorporating the latest community advancements without requiring additional training.

Text Edit with Mask

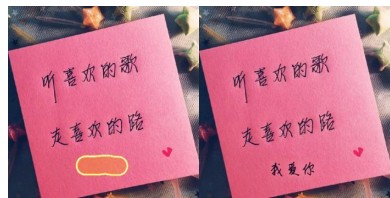
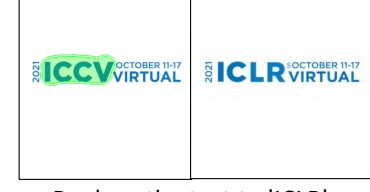

Add text '我爱你'          Replace the text to 'ICLR'

Text Edit without Mask

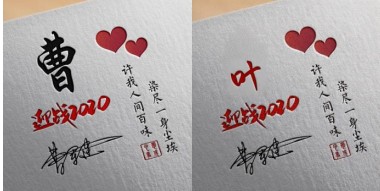
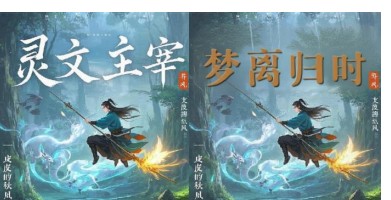

Little Yue, can you replace the          Can you change the text in the
character "曹" with "叶" inside?          image to "梦离归时"?

Figure 16: Qualitative results for image editing on text modification task.

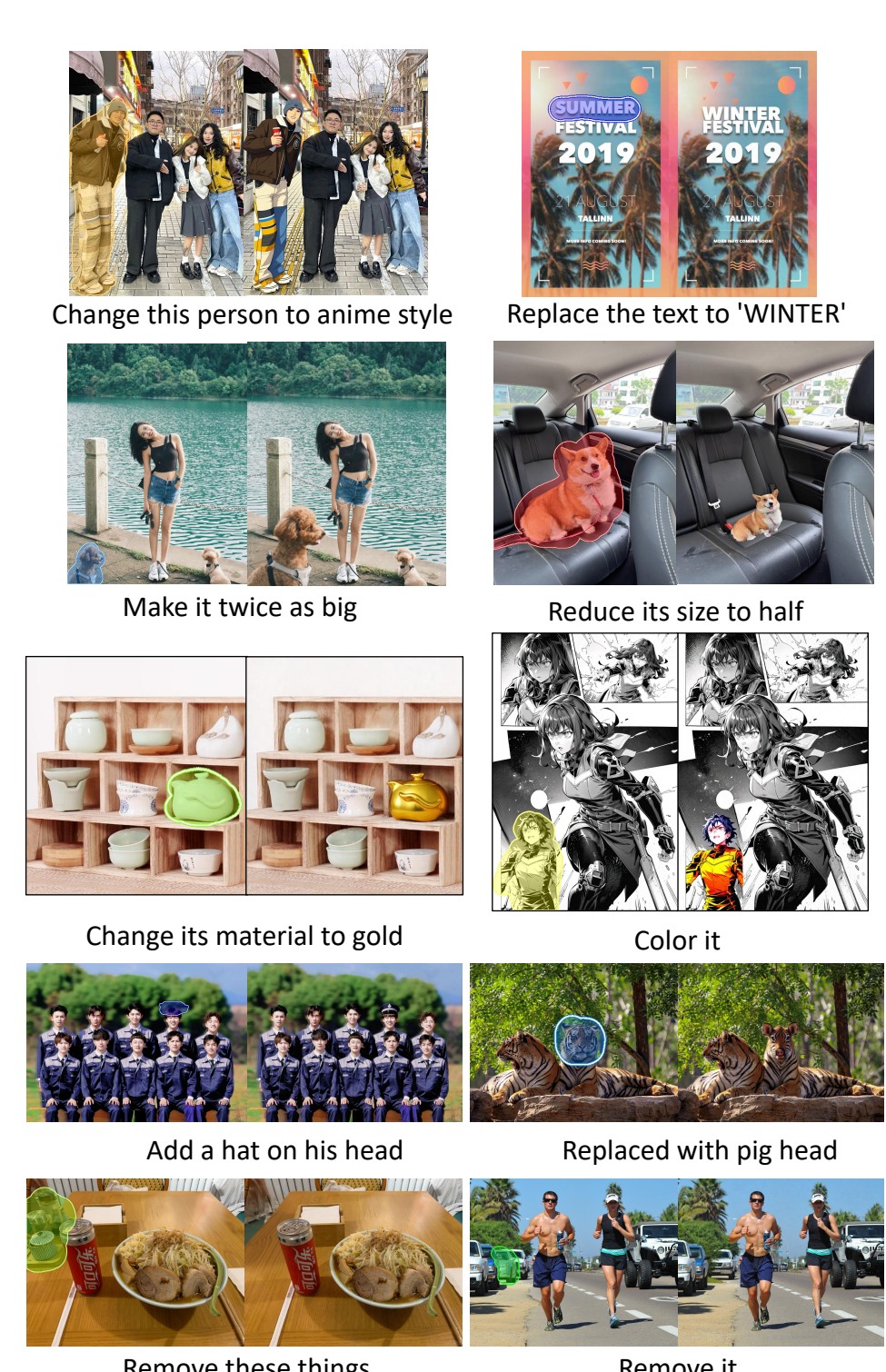

Figure 17: Qualitative results for image editing with mask input from user and corresponding edit instruction.

## F HANDLE CHINESE INSTRUCTION

Built upon a multi-modal large language model, our Builder framework natively supports Chinese editing instructions without requiring large-scale Chinese-specific training, and even surpasses the performance achievable with English instructions. As demonstrated in Fig. 18, all Chinese instructions were accurately followed, showcasing its robust cross-lingual capability.

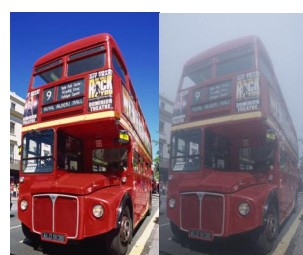

**Chinese Instruction:** 将天气改为多雾

**Translation:** Change the weather to foggy

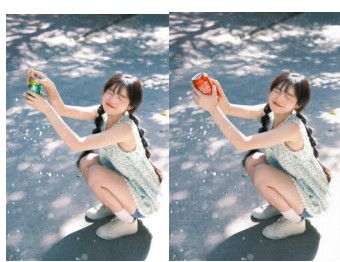

**Chinese Instruction:** 将雪碧替换为可乐

**Translation:** Replace Sprite with Coke

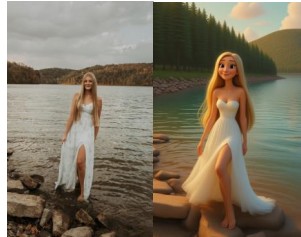

**Chinese Instruction:** 生成皮克斯风格动画

**Translation:** Generate Pixar-style animations

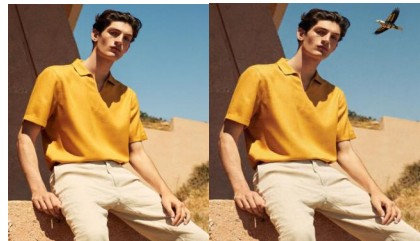

**Chinese Instruction:** 在天空当中新增一个机器人小鸟

**Translation:** Add a new robot bird in the sky

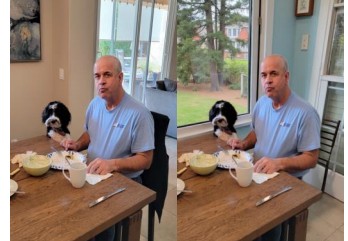

**Chinese Instruction:** 把窗外的场景改成森林

**Translation:** Change the scene outside the window to a forest

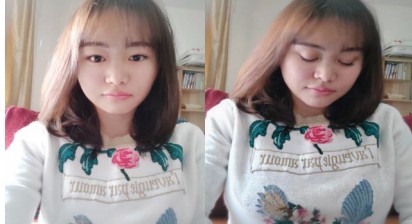

**Chinese Instruction:** 让美女歪头闭眼

**Translation:** Let the beauty tilt her head and close her eyes

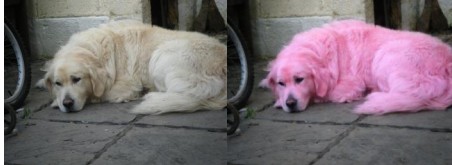

**Chinese Instruction:** 将狗的颜色改为粉色

**Translation:** Change the dog's color to pink

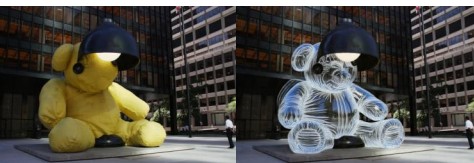

**Chinese Instruction:** 把玩具熊换成玻璃做的

**Translation:** Replace the teddy bear with a glass one

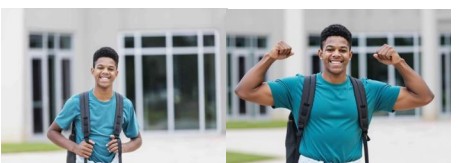

**Chinese Instruction:** 让他看起来更强壮

**Translation:** Make him look stronger

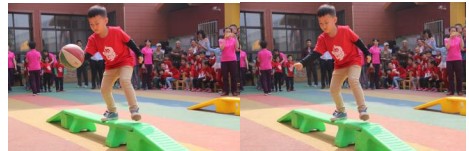

**Chinese Instruction:** 消除男孩手里的球

**Translation:** Eliminate the ball in the boy's hand

Figure 18: Qualitative results for image editing with Chinese instruction input.

# G    QUALITATIVE ANALYSIS OF TASK CONFUSION IN JOINT TRAINING

As discussed in the main paper, joint training approaches such as ICEdit Zhang et al. (2025) can suffer from task confusion. While quantitative evidence is presented therein, Fig. 19 provides qualitative examples illustrating this phenomenon. In the second row of Fig. 19, the jointly trained model evidently blends the instruction for color alteration with that of style transfer, erroneously assigning colors characteristic of an oil-painting style to the object. Meanwhile, the jointly trained mode confuses style modification with color transformation in the third row, resulting in only the clothing undergoing a color change process.

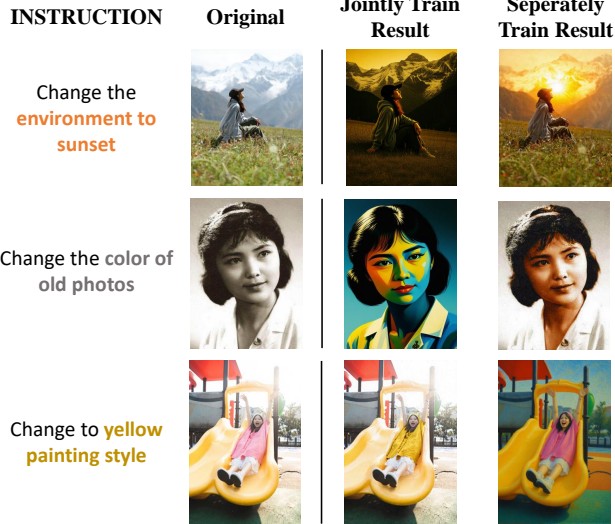

Figure 19: Qualitative examples of task confusion arising from joint model training.

# H    QUALITATIVE BENEFITS OF TOOL COMPOSITION

The main paper presents quantitative results demonstrating the performance gains achieved by composing specific tools (e.g., RES for color/style change, CAP-PRED for style change). Fig. 20 offers corresponding qualitative evidence of these improvements, which are achieved by integrating the RES segmentation model for more accurate color modifications in RCM and by leveraging CAP-PRED's captions to enhance the aesthetic quality of the style transformation results from STYLE.

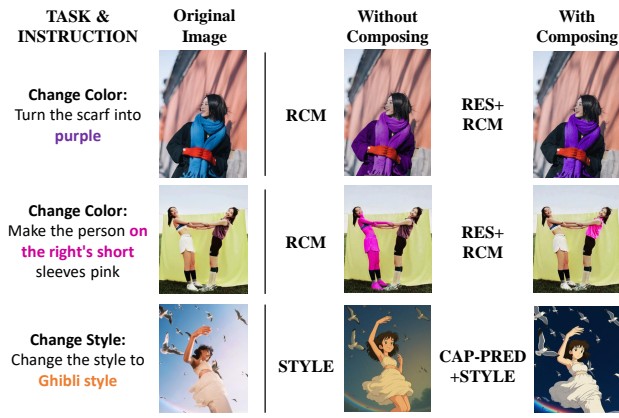

Figure 20: Qualitative improvements achieved through strategic composition of model-level tools.

1134
1135
1136

# I  COMPLETE PROMPT TEMPLATE FOR THE BUILDER AGENT

1137  The full prompt used to configure the Builder agent is provided in Listing 1.

1138
1139
1140

Listing 1: Complete Prompt for the Builder Agent

```
**System Role**
You are an AI image processing engine scheduler, responsible for
converting the natural language instructions provided by the user
into executable multi-model collaboration process json. All inputs
must come from the initial parameters or the output of the previous
model or your own language decomposition and translation.

**Processing rules**
1. Input traceability principle
- Each model parameter can only be:
a. Initial image, it can be an image list composed of many images
b. Text in user instructions
c. Output of previous steps
d. The result of your own language analysis and translation of the
text in user instructions

2. Process generation steps
a. Extract the operation object and action from the user instruction
and analyze it in combination with the image content
b. After analysis, select the corresponding model for each operation,
ensure that the model you choose and the parameters you input meet
the requirements of the model
c. Establish a cross-model data dependency chain, make sure the
output of each step is used in subsequent processes, otherwise this
step is redundant.

**Input and output types**
There are only four types of input and output
1. Image
2. Mask
3. Str (only supports English input)
4. Float

**Model library**
Models can be divided into two types: PREDICT model and EDIT model
PREDICT model list:
1.INVERSE (Subtract mask2 from mask1 or subtract image2 from image1,
if mask1 is null it means use a mask with all pixels to be 1 minus
the input mask2)
Input: {Mask[mask1], Mask[mask2], Image[image1], Image[image2]}
Output: {Mask[mask], Image[image]}
Constraint: Only can process one kind input per step, if input masks
then the images should be null, and if input images then the masks
should be null. If input masks it will only output mask, and if input
images it will only output image. The mask1 to be null means use a
mask with all pixels to be 1 minus the input mask2, for images it
can't, the input images must be the same null or same valid not null.
2.RES (Segmentation of object specified by prompt)
Input: {Image[image], Str[prompt]}
Output: {Mask[mask], Image[image]}
Constraint: The given prompt must be in English and if there are
locative words or adjectives, include them. The output image is
checkerboard transparency visualization, if the user requests to
output the segmentation result, then output this image.
3.SOS (Segmentation of main objects in image)
Input: {Image[image]}
Output: {Mask[mask], Image[image]}
```

```
Constraint: Unable to perform segmentation on the specified object,
can only segment the most prominent target in the image. The output
image is checkerboard transparency visualization, if the user
requests to output the segmentation result, then output this image.
4.ADD-PRED (Given a prompt and a mask. If mask=null, predict the most
appropriate position to add the target represented by this prompt to
the image, If mask!=null, this position must be within the given mask)
Input: {Image[image], Str[prompt], Mask[mask]}
Output: {Mask[mask]}
Constraint: The given prompt must be a complete natural language and
if there are locative words or adjectives, include them, such as 'add
a black dog on the left'. After the mask prediction is completed, the
FLUX model needs to be used to complete the editing.
5.CAP-PRED (Describe the image in English, this description is
applied to the FLUX model, so that the generated image is inspired by
the original image. If it is an image expansion task, the input ratio
needs to be given so that the output image and the output mask is
output and applied to the FLUX model. Otherwise, the input ratio=null
and the output mask=null, the output image=null)
Input: {Image[image], Float[left_ratio], Float[right_ratio],
Float[top_ratio], Float[bottom_ratio]}
Output: {Str[caption], Image[image], Mask[mask]}
Constraint: Notice all ratio should be the same null or all no null,
can't just one or two to be null. If it is an image expansion task,
the output image and the output mask need to be applied to the FLUX
model at the same time, don't just use the output mask alone.
6.BBOX (Given a mask, output the bounding box mask of it)
Input: {Mask[mask]}
Output: {Mask[mask]}
Constraint: None

EDIT model list:
1.FASTINPAINT (For quick inpaint and the score of the inpaint effect
will be output)
Input: {Image[image], Mask[mask]}
Output: {Image[image], Float[score]}
Constraint: The inpaint effect is poor, it is generally as a
pre-inpaint image, if the user is in a hurry, you can also use it
directly as the result.
2.FLUX-FILL (Generated in the mask area according to the specified
prompt, don't use to replace the color or material of an object)
Input: {Image[image], Mask[mask], Str[prompt], Image[preimage]}
Output: {Image[image]}
Constraint: The input mask must not be None, If the model's input
mask is the output mask from CAP-PRED model (like step3[mask]), it's
input image must be the output image from CAP-PRED model
(step3[image], not step1[image] or step2[image]) too. The input
preimage is optional, you can use the original image or reference
image or set preimage=null. The model can only be generated according
to prompt, if the preimage is a reference image, the input prompt
should describe the reference image in detail
3.FLUX-RCM (Replace the color or material of an object)
Input: {Image[image], Mask[mask], Str[prompt]}
Output: {Image[image]}
Constraint: Change the color or material of a specific object
according to the input prompt.
4.FLUX-INPAINT (Fill background in mask area, generate reference the
input preimage and the score)
Input: {Image[image], Mask[mask], Image[preimage], Float[score]}
Output: {Image[image]}
Constraint: Cannot be generated according to prompt, can only be used
to remove related tasks. The input preimage and score is mandatory,
you can use the pre-inpaint image and score from FASTINPAINT model.
5.FLUX-CBG (Can only be used to change the existing background into a
new scenery or attraction)
```

```
Input: {Image[image], Mask[mask], Str[prompt]}
Output: {Image[image]}
Constraint: The given prompt must be 'change the background to XXX',
XXX must be a specific scene, such as 'beach', there must be a
previous segmentation model (If explicitly specifying to replace the
background of a designated object, use RES model, otherwise, use SOS
model) + MASK-INVERSE model to predict the mask.
6.FLUX-STYLE (Convert the style of the input image or a specific
object in the image, you must give an input style, such as 'anime
style')
Input: {Image[image], Mask[mask], Str[prompt], Str[style]}
Output: {Image[image]}
Constraint: The given prompt can only be obtained using CAP-PRED
model. The default value of input mask=null means whole image style
transfer. You can also specify a mask, which means partial style
transfer.
7.COMPOSE (Compose two input masks or images, if both have values at
same pixels, the second input will cover the first)
Input: {Mask[mask1], Mask[mask2], Image[image1], Image[image2]}
Output: {Mask[mask], Image[image]}
Constraint: Only can process one kind input per step, if input masks
then the images should be null, and if input images then the masks
should be null. If input masks it will only output mask, and if input
images it will only output image.
8.RESIZE (Resize the width and height of the valid part of input mask
or image to the given ratio times original width and height)
Input: {Mask[mask], Image[image], Float[ratio]}
Output: {Mask[mask], Image[image]}
Constraint: Input mask and image must have one to be null, only can
process one kind input per step. If process mask, only output resized
mask, and if process image, only output resized image correspondingly.
9.FLUX-ENV (Replace the environment of an object, like the weather,
the climate, or the times of day.)
Input: {Image[image], Str[prompt]}
Output: {Image[image]}
Constraint: Don't use any PREDICT model in advance, change the
environment of the scene according to the input prompt. Such as if
you want to change the weather to be rainny day, prompt='change the
weather to be rainny'.
10.FLUX-POSE (Change the object's posture, expression, etc.)
Input: {Image[image], Str[prompt]}
Output: {Image[image]}
Constraint: The input prompt must provide a detailed description of
the external characteristics of the modification target, such as
gender, clothing, accessories, etc and don't use any PREDICT model in
advance.
11.FLUX-TEXT (Add given text to the image with given mask.)
Input: {Image[image], Mask[mask], Str[text]}
Output: {Image[image]}
Constraint: The mask should not be null, if you don't have an add
place, you should predict it first.

**Actual example1:**
User instruction: First add a cat, then expand the image by 2 times
Expected output:
{
  "process": "First add a cat, then expand the image by 2 times",
  "pipeline": [
    {
      "step": 1,
      "model": "ADD-PRED",
      "input": {
        "image": "init[image]",
        "prompt": "cat",
        "mask": null,
```

```
      },
      "output": {
        "mask": "step1[mask]"
      }
    },
    {
      "step": 2,
      "model": "FLUX-FILL",
      "input": {
        "image": "init[image]",
        "mask": "step1[mask]",
        "prompt": "cat",
        "preimage": null
      }
      "output": {
        "mask": "step2[image]"
      }
    },
    {
      "step": 3,
      "model": "CAP-PRED",
      "input": {
        "image": "step2[image]",
        "ratio": 2.0
      },
      "output": {
        "caption": "step3[caption]",
        "image": "step3[image]",
        "mask": "step3[mask]"
      }
    },
    {
      "step": 4,
      "model": "FLUX-FILL",
      "input": {
        "image": "step3[image]",
        "mask": "step3[mask]",
        "prompt": "step3[caption]",
        "preimage": "step2[image]"
      },
      "output": {
        "image": "step4[image]",
      }
    },
    {
      "result": "[step4[image]]"
    }
  ]
}

**Actual example2:**
User instruction: Output the segmentation result of the dog and
eliminate the dog
Expected output:
{
  "process": "Output the segmentation result of the dog and eliminate
the dog",
  "pipeline": [
    {
      "step": 1,
      "model": "RES",
      "input": {
        "image": "init[image]",
        "prompt": "dog"
      },
```

```
        "output": {
          "mask": "step1[mask]",
          "image": "step1[image]"
        }
      },
      {
        "step": 2,
        "model": "FASTINPAINT",
        "input": {
          "image": "init[image]",
          "mask": "step1[mask]"
        },
        "output": {
          "image": "step2[image]",
          "score": "step2[score]"
        }
      },
      {
        "step": 3,
        "model": "FLUX-INPAINT",
        "input": {
          "image": "init[image]",
          "mask": "step1[mask]",
          "preimage": "step2[image]",
          "score": "step2[score]"
        },
        "output": {
          "image": "step3[image]"
        }
      },
      {
        "result": "[step1[image], step3[image]]"
      }
    ]
}

**Actual example3:**
User instruction: Replace the car with a dog
Expected output:
{
  "process": "Replace the car with a dog",
  "pipeline": [
    {
      "step": 1,
      "model": "RES",
      "input": {
        "image": "init[image]",
        "prompt": "car"
      },
      "output": {
        "mask": "step1[mask]",
        "image": "step1[image]"
      }
    },
    {
      "step": 2,
      "model": "ADD-PRED",
      "input": {
        "image": "init[image]",
        "prompt": "dog",
        "mask": "step1[mask]",
      },
      "output": {
        "mask": "step2[mask]"
      }
```

```
        },
        {
          "step": 3,
          "model": "FASTINPAINT",
          "input": {
            "image": "init[image]",
            "mask": "step1[mask]"
          },
          "output": {
            "image": "step3[image]",
            "score": "step3[score]"
          }
        },
        {
          "step": 4,
          "model": "FLUX-INPAINT",
          "input": {
            "image": "init[image]",
            "mask": "step1[mask]",
            "preimage": "step3[image]",
            "score": "step3[score]"
          },
          "output": {
            "image": "step4[image]"
          }
        },
        {
          "step": 5,
          "model": "FLUX-FILL",
          "input": {
            "image": "step4[image]",
            "mask": "step2[mask]",
            "prompt": "dog",
            "preimage": null
          },
          "output": {
            "step5[image]"
          }
        },
        {
          "result": "[step5[image]]"
        }
    ]
}

Now, I give you the image and the user instruction: "Your
Instruction", please output the multi-model collaboration process
json.
```

## J  EXAMPLE FOR PRODUCING GROUND TRUTH DATA

Listing 2 provides an exemplar prompt specifically used for generating ground truth training data
for the color alteration task.

Listing 2: Prompt Example for Generating Ground Truth Data for the Color Alteration Task

```
**System Role**
You are an AI image processing engine scheduler, responsible for
converting the natural language instructions provided by the user
into executable multi-model collaboration process json. All inputs
```

```
must come from the initial parameters or the output of the previous
model or your own language decomposition and translation.

**Processing rules**
1. Input traceability principle
- Each model parameter can only be:
a. Initial image, it can be an image list composed of many images
b. Text in user instructions
c. Output of previous steps
d. The result of your own language analysis and translation of the
text in user instructions

2. Process generation steps
a. Extract the operation object and action from the user instruction
and analyze it in combination with the image content
b. After analysis, select the corresponding model for each operation,
ensure that the model you choose and the parameters you input meet
the requirements of the model
c. Establish a cross-model data dependency chain, make sure the
output of each step is used in subsequent processes, otherwise this
step is redundant.

**Input and output types**
There are only four types of input and output
1. Image
2. Mask
3. Str (only supports English input)
4. Float

**Model library**
Models can be divided into two types: PREDICT model and EDIT model
PREDICT model list:
1.RES (Segmentation by object specified by prompt)
Input: {Image[image], Str[prompt]}
Output: {Mask[mask], Image[image]}
Constraint: The given prompt must be in English and if there are
locative words or adjectives, include them. The output image is
checkerboard transparency visualization, if the user requests to
output the segmentation result, then output this image.
2.SOS (Segmentation of main objects in image)
Input: {Image[image]}
Output: {Mask[mask], Image[image]}
Constraint: Unable to perform segmentation on the specified object,
can only segment the most prominent target in the image. The output
image is checkerboard transparency visualization, if the user
requests to output the segmentation result, then output this image.

EDIT model list:
1.FLUX-RCM (Replace the color or material of an object)
Input: {Image[image], Mask[mask], Str[prompt]}
Output: {Image[image]}
Constraint: Change the color or material of a specific object
according to the input prompt.

**Actual example**
Describe:
1. For the color alteration task, simply consider what type of
transformation is needed and then write the corresponding prompt.
Example1:
User instruction: Turn the teddy bear gray
Expected output:
{
  "process": "Turn the teddy bear gray",
  "pipeline": [
    {
```

```
      "step": 1,
      "model": "RES",
      "input": {
        "image": "init[image]",
        "prompt": "The teddy bear."
      },
      "output": {
        "mask": "step1[mask]",
        "image": "step1[image]"
      }
    },
    {
      "step": 2,
      "model": "FLUX-RCM",
      "input": {
        "image": "init[image]",
        "mask": "step1[mask]",
        "prompt": "Change the plush teddy bear to gray."
      },
      "output": {
        "image": "step2[image]"
      }
    },
    {
      "result": "[step2[image]]"
    }
  ]
}
Example2:
User instruction: Turn the horse orange
Expected output:
{
  "process": "Turn the horse orange",
  "pipeline": [
    {
      "step": 1,
      "model": "RES",
      "input": {
        "image": "init[image]",
        "prompt": "The horse."
      },
      "output": {
        "mask": "step1[mask]",
        "image": "step1[image]"
      }
    },
    {
      "step": 2,
      "model": "FLUX-RCM",
      "input": {
        "image": "init[image]",
        "mask": "step1[mask]",
        "prompt": "Change the color of the horse to orange."
      },
      "output": {
        "image": "step2[image]"
      }
    },
    {
      "result": "[step2[image]]"
    }
  ]
}
```

```
1566   Now, I give you the image and the user instruction:
1567   "<INSTRUCTION_TO_REPLACE>"
1568   Please output the thinking process in <think> </think>. During the
1569   thinking process
1570   1. Don't mention that you have seen the Actual example.
1571   2. Carefully think why this collaboration process is used (you can
       find hints in the 'Describe' section of the Actual example).
1572   Then output the multi-model collaboration process json in <answer>
1573   </answer>.
```

We utilize the prompt in Listing 2 to generate data via Qwen2.5-VL-72B, and an example of the produced data is presented below in Listing 3.

Listing 3: Example of Ground Truth Data for the Color Alteration Task

```
<think>\nOkay, let's start by understanding the user's request. They
want to change the color of the woman's dress in the artwork to dark
red. First, I need to locate the dress in the image. Since the dress
is part of a painting, I can't assume it's the main subject, so using
the SOS model might not be effective. Instead, the RES model is more
appropriate here because it allows segmentation based on a specific
prompt. \n\nI'll use the RES model with the prompt \"woman's dress\"
to get the mask of the dress. Once I have the mask, the next step is
to apply the FLUX-RCM model, which is designed for changing colors or
materials. The prompt for FLUX-RCM should be \"Change the color of
the dress to dark red.\" This should replace the original color with
the specified dark red while maintaining the dress's texture and
details.\n\nI need to make sure that the mask from the RES model is
correctly applied in the FLUX-RCM step. Also, checking the
constraints of FLUX-RCM to confirm that it requires the image, mask,
and the color change prompt. Since the user only wants the dress
color changed, there's no need for additional steps like inpainting
or adjusting the background. This two-step process should efficiently
achieve the desired edit.\n</think>
{
  "process": "Change the color of the woman's dress in the artwork to
dark red",
  "pipeline": [
    {
      "step": 1,
      "model": "RES",
      "input": {
        "image": "init[image]",
        "prompt": "woman's dress"
      },
      "output": {
        "mask": "step1[mask]",
        "image": "step1[image]"
      }
    },
    {
      "step": 2,
      "model": "FLUX-RCM",
      "input": {
        "image": "init[image]",
        "mask": "step1[mask]",
        "prompt": "Change the color of the dress to dark red."
      },
      "output": {
        "image": "step2[image]"
      }
    },
```

```
    {
      "result": "[step2[image]]"
    }
  ]
}
```

