# OpenReview forum: "Lego-Edit: A General Image Editing Framework with Model-Level Bricks and MLLM Builder"
_ICLR.cc/2026/Conference — Submitted to ICLR 2026_

### Official Review · Reviewer_ozbk · 2025-10-16

**Soundness:** 3
**Presentation:** 2
**Contribution:** 2
**Rating:** 4
**Confidence:** 4

**Summary:**

This paper introduces Lego-Edit, a modular framework for instruction-based image editing. The system consists of a Builder (a multimodal large language model fine-tuned with reinforcement learning) that organizes a library of model-level tools, Bricks, such as segmentation, inpainting, and style-transfer modules. A three-stage progressive RL scheme, supervised fine-tuning, ground-truth-based RL, and GT-free RL guided by a critic mode, gradually improves the Builder’s reasoning and tool-composition ability. Experiments on GEdit-Bench and ImgEdit-Bench show performance gains over strong baselines (e.g., BAGEL, Step1X-Edit, UniWorld-V1). The system also demonstrates zero-shot generalization and the ability to integrate new tools without retraining.

**Strengths:**

- The paper introduces a well-motivated modular architecture that addresses the limited flexibility of end-to-end image-editing models.
- The three-stage reinforcement-learning pipeline is a strong design choice that contributes to improved reasoning and generalization.
- Experimental results are extensive and include both quantitative benchmarks and visual demonstrations that convincingly illustrate the system’s strengths.
- The implementation details and reproducibility statement are thorough and transparent.
- The framework shows promising potential for extensibility and future multimodal applications.

**Weaknesses:**

- The contribution is primarily engineering-focused. The paper combines known techniques, MLLM agents, RL fine-tuning, LoRA adapters, into a unified system rather than introducing a new theoretical or algorithmic idea. While the integration is impressive, it offers limited conceptual insight into why or how the components interact optimally.
- The complexity of the architecture (Builder, Executor, Bricks, Critic) makes it difficult to interpret which design choices drive performance gains. More controlled ablations would clarify the marginal value of each component.
- The critic model used in the final RL stage plays an important conceptual role but is evaluated mostly qualitatively; quantitative ablations or alternative critic formulations would strengthen the argument.
- The writing and structure could be improved to highlight the core intuition behind the framework before diving into system specifics.
- The evaluation scope is limited to existing benchmarks. Although results are strong, the paper would benefit from evidence of user studies or real-world use cases demonstrating the practicality of Lego-Edit in diverse conditions.
- Some important limitations and failure modes are only briefly mentioned (e.g., potential tool conflicts, scalability bottlenecks) and deserve more explicit discussion.
- Runtime analysis is incomplete. The paper reports latency only relative to BAGEL but does not compare against other strong baselines such as Step1X-Edit, OmniGen2, or FLUX.1 Kontext. A broader efficiency analysis would help position Lego-Edit’s computational trade-offs relative to state-of-the-art models.

**Questions:**

1. How dependent is Lego-Edit’s performance on the chosen MLLM backbone (MiMo-VL-7B)? Could smaller or publicly available models achieve comparable coordination ability?
2. Could the authors report quantitative results comparing Stage 2 (GT-based RL) and Stage 3 (critic-based RL) to show how much each stage improves reasoning or compositional success rate?
3. How does the framework scale when the number of available tools increases or when workflows become more complex? Are there computational or latency trade-offs?
4. What are the typical failure modes? For example, does the Builder sometimes misinterpret instructions, or do tool boundaries introduce visual artifacts?
5. Since the paper is positioned in “applications to multiple modalities,” how feasible would it be to extend this framework to non-visual domains such as audio or video editing?
6. Have the authors evaluated how easily new users can add tools or control the Builder’s behavior without retraining? Understanding the system’s real-world maintainability could increase its practical impact.
7. Beyond the single latency result against BAGEL, could the authors provide runtime or computational cost comparisons with other leading systems such as Step1X-Edit, OmniGen2, or FLUX.1 Kontext? Such information would clarify Lego-Edit’s practical efficiency.

---

> ### Author Response · Authors · 2025-11-20
>
> We sincerely thank you for the careful reading and thoughtful comments. We address each point in detail below.
>
> ---
>
> > **The contribution is primarily engineering-focused. The paper combines known techniques, MLLM agents, RL fine-tuning, LoRA adapters, into a unified system rather than introducing a new theoretical or algorithmic idea. While the integration is impressive, it offers limited conceptual insight into why or how the components interact optimally.**
>
> Thank you for raising this important point. Although our system employs several existing components, our contributions go beyond a simple combination of techniques. We highlight two core innovations:
>
> 1. Fine-grained tool design instead of coarse task-level workflows. Prior methods such as ComfyMind (NeurIPS 2025) rely on manually designed, task-level workflows. In contrast, we design fine-grained tools that can be flexibly composed. This is the foundation that allows the Builder to generalize to new functionalities and even support arbitrarily extended capabilities.
>
> 2. A reinforcement learning framework that teaches the Builder how to think and compose tools. Such fine-grained tools impose a significant burden on an untrained MLLM. Off-the-shelf models cannot perform reliable tool composition. Our tailored reinforcement learning (with ground truth in Stage 2 and evaluator feedback in Stage 3) strengthens the Builder’s reasoning ability while preserving its ability to generalize to unseen tools.
>
> With these two innovations combined, our system achieves performance far beyond ComfyMind and even surpasses several strong end-to-end models. Moreover, this design substantially expands the set of tasks the system can handle, enabling fairly free editing. For these reasons, we believe the conceptual and practical contributions extend well beyond engineering integration.
>
> ---
>
> > **The evaluation scope is limited to existing benchmarks. Although results are strong, the paper would benefit from evidence of user studies or real-world use cases demonstrating the practicality of Lego-Edit in diverse conditions.**
>
> Sorry for the confusion. Due to space limitations, we included more real-world usage examples in the supplementary material, including additional comparisons (Section B.3), real-world text editing (Section E), and mask-based editing results (Section D). These examples demonstrate the practicality of Lego-Edit under real usage conditions. And we have also conducted human evaluation on some volunteers, the result is reported in Section B.1 of the supplementary material which demonstrates our superior performance.
>
> ---
>
> > **How dependent is Lego-Edit’s performance on the chosen MLLM backbone (MiMo-VL-7B)? Could smaller or publicly available models achieve comparable coordination ability?**
>
> We appreciate this question. MiMo-VL-7B was chosen because it was the only small-scale multimodal reasoning model available at the time. Recently, Qwen3-VL-2B was released, and we trained Lego-Edit using this smaller backbone as well. The extended results are shown below:
>
> | Agent Model | Simple %Pass | Complex %Pass | G_SC↑ | G_PQ↑ | G_O↑ |
> |-------------|--------------|---------------|-------|-------|------|
> | GPT-4o | 75.2 | 52.1 | 4.28 | 6.53 | 5.10 |
> | MiMo-VL-7B-Base | 56.8 | 44.9 | 3.11 | 4.66 | 3.59 |
> | MiMo-SFT | 75.7 | 53.8 | 4.39 | 7.01 | 5.22 |
> | MiMo-RL w/ GT | 100 | 81.9 | 5.49 | 7.35 | 6.16 |
> | MiMo-RL w/o GT | 100 | 99.1 | 6.45 | 7.45 | 6.88 |
> | Qwen3-VL-2B-Base | 49.6 | 39.8 | 2.74 | 6.01 | 3.12 |
> | Qwen3-SFT | 68.3 | 47.1 | 4.17 | 6.58 | 4.99 |
> | Qwen3-RL w/ GT | 94.8 | 73.7 | 5.33 | 7.32 | 6.09 |
> | Qwen3-RL w/o GT | 100 | 98.5 | 6.40 | 7.41 | 6.85 |
>
> Although the Qwen3-VL-2B backbone has weaker initial capability, following our multi-stage training process significantly boosts performance to be comparable to the MiMo-based version. This indicates that the effectiveness of Lego-Edit comes primarily from our data construction and training pipeline, rather than from a specific backbone.
>
> ---
>
> > **Could the authors report quantitative results comparing Stage 2 (GT-based RL) and Stage 3 (critic-based RL) to show how much each stage improves reasoning or compositional success rate?**
>
> Thank you for the question. The paper reports the improvements brought by both stages (Table 5), including execution success rate and performance on the image editing benchmarks. In the previous response, we also provided additional results from Qwen3-VL-2B following the same training pipeline.

---

> > ### Author Response · Authors · 2025-11-20
> >
> > ---
> >
> > > **How does the framework scale when the number of available tools increases or when workflows become more complex? Are there computational or latency trade-offs?**
> >
> > Thank you for raising this. Our framework does not introduce notable computational or latency trade-offs. Even for highly complex instructions, the Builder executes the full workflow faithfully. For example, when given the instruction to enlarge an image 20 times sequentially (each by 1.25×), the Builder correctly outputs a 40-step workflow, enabling a zoom-out video-like effect when concatenating intermediate outputs.
> >
> > In practice, most user instructions require fewer than four steps. Adding more tools only requires inserting their descriptions into the prompt and therefore has little impact on chain-of-thought length or inference efficiency. The runtime statistics are provided in the response to the final question.
> >
> > ---
> >
> > > **What are the typical failure modes? For example, does the Builder sometimes misinterpret instructions, or do tool boundaries introduce visual artifacts?**
> >
> > Thank you for the question. We have added a section to the paper discussing failure modes (Section 5). In our testing, instruction misinterpretation is rare, and tools seldom introduce visual artifacts.
> >
> > ---
> >
> > > **Since the paper is positioned in “applications to multiple modalities,” how feasible would it be to extend this framework to non-visual domains such as audio or video editing?**
> >
> > This is an excellent idea. We believe such extensions are highly feasible.
> > For inputs, audio can be seamlessly incorporated by introducing an automatic audio-to-text model as a callable tool. Similar to the mask-input mechanism demonstrated in our supplementary material (Section D), the Builder can be prompted to recognize the presence of an audio input and invoke the conversion tool accordingly. Video input can be handled in a similar manner, as many recent MLLMs already provide video understanding capabilities. Moreover, emerging unified multimodal foundation models such as Qwen-Omni, which jointly process text, images, video, and audio, offer an alternative backbone choice that could further enhance cross-modal support and open up broader possibilities for future extensions.
> > For outputs, extending the toolset is straightforward; we only need to add tool descriptions for video or audio editing to the prompt, and Builder can reason over them.

---

> > > ### Author Response · Authors · 2025-11-20
> > >
> > > ---
> > >
> > > > **Have the authors evaluated how easily new users can add tools or control the Builder’s behavior without retraining? Understanding the system’s real-world maintainability could increase its practical impact.**
> > >
> > > Thank you for pointing out this important practical consideration. Regarding adding tools or controlling Builder behavior, our experiments confirm that new tools can be introduced simply by adding their descriptions to the prompt. Moreover, for new users, the implementation is intentionally kept simple and transparent. Adding a new tool only requires defining a small inference function with proper inputs and outputs, registering its name in a dictionary, and inserting a short capability description into the Builder prompt. Builder will then autonomously determine when and how to invoke it. For example, enabling mask inputs requires only adding a short description indicating that the user provides a mask without any retraining.
> > >
> > > We emphasize maintainability in both system design and experimentation. We replace all FLUX tools with Qwen-based counterparts within a single day, which illustrates that the system should be easy to extend for new users, and such replacement yields noticeable performance gains, confirming the maintainability of our framework. Moreover, as discussed in the newly added Section 5 on failure cases, a modest enhancement of the capability for segmenting the hour hand enables the system to correctly address the case of changing time for the clock, further demonstrating the maintainability of our framework.
> > >
> > > These demonstrate that Lego-Edit is lightweight, easy to maintain, and straightforward to extend in real-world scenarios.
> > >
> > > ---
> > >
> > > > **Beyond the single latency result against BAGEL, could the authors provide runtime or computational cost comparisons with other leading systems such as Step1X-Edit, OmniGen2, or FLUX.1 Kontext? Such information would clarify Lego-Edit’s practical efficiency.**
> > >
> > > Thank you for the suggestion. We conducted additional latency evaluations and report the results below:
> > >
> > > - **Bagel:** 25.0 s
> > > - **Step1X-Edit:** 21.3 s
> > > - **Omnigen2:** 22.6 s
> > > - **FLUX-Kontext:** 7.4 s
> > >
> > > - **Our MiMo Builder:** 3.5 s + slowest Tool 2.7 s → **Total ≈ 6.9 s**
> > > - **Our Qwen3-VL-2B Builder:** 1.9 s + slowest Tool 2.7 s → **Total ≈ 5.2 s**
> > >
> > > These results show that Lego-Edit achieves competitive pipeline latency compared to existing state-of-the-art systems.
> > >
> > > ---
> > >
> > > Once again, we thank you for the insightful and constructive comments. We hope our revisions and clarifications adequately address your concerns, and we would be grateful for your positive assessment in the final decision.

---

> > > > ### Comment · Reviewer_ozbk · 2025-11-25
> > > >
> > > > Thank you for your effort during the rebuttal. All my concerns have been addressed, and I appreciate that some updates are already included in the revised version. Please ensure that all discussed points are added to the final version, e.g., the latency comparison with other methods.
> > > >
> > > > I am increasing my rating to 8.

---

> > > > > ### Author Response · Authors · 2025-11-26
> > > > >
> > > > > Thank you very much for your careful reconsideration of our response. We sincerely appreciate your positive feedback. All revisions discussed during the rebuttal, including the latency comparison with other methods, have been incorporated into the manuscript and highlighted in blue for clarity. The updated version has already been submitted.
> > > > >
> > > > > Thank you again for your constructive comments and for increasing the rating. Your feedback has been invaluable in improving the quality of our work.

---

### Official Review · Reviewer_4Y2L · 2025-10-26

**Soundness:** 2
**Presentation:** 3
**Contribution:** 2
**Rating:** 4
**Confidence:** 3

**Summary:**

This paper targets on the problem that though have somewhat good editing performance, the current instruction-based image editing model always fail to generalize to unknown instructions. They propose a method called Lego-Edit, it finetunes a Multi-modal Large Language Model (MLLM) by reinforcement learning, enable it to coordinate model-level editing tools. In the MLLM part, it three-stage progressive reinforcement learning training strategy to make the MLLM more adapted to the current task. Extensive experiment results prove the effectiveness of the proposed method.

**Strengths:**

1. The paper is in a good organization, the idea is easy to understand.
2. The experiment section is good, with abundant results and clear explanation. So I think the results are convincing.

**Weaknesses:**

1. Some typos, eg. Line 182, fine-graine. Please check the whole paper.
2. If I do not mis-understand, since you incorporate more models, can you have some discussions about the train / inference efficiency of the proposed method?
3. I am willing to see more discussions about the novelty of the proposed method, since dividing one editing task to multiple different tasks is not interesting enough I think.
4. I also want the authors to have more discussions about the limitation and future work of the work.

**Questions:**

See weaknesses. I think this paper is ok, but we can have more discussions during the rebuttal period. If my concerns are solved, I will be happy to raise the score. I hope the discussions can also bring about some new insights.

---

> ### Author Response · Authors · 2025-11-20
>
> We sincerely thank you for the careful reading and constructive feedback. We address each comment in detail below.
>
> ---
>
> > **Some typos, eg. Line 182, fine-graine. Please check the whole paper.**
>
> Thank you for pointing this out. We have carefully re-checked the entire manuscript and corrected all typos, including the one noted at Line 182.
>
> ---
>
> > **If I do not mis-understand, since you incorporate more models, can you have some discussions about the train / inference efficiency of the proposed method?**
>
> We appreciate your question. Below we provide a detailed discussion of both training and inference efficiency.
>
> For the training efficiency, the Builder is highly efficient to train, on 8×H20 GPUs, full training requires only about five days. Once the Builder is trained, newly added tools require no additional training of the Builder, significantly reducing maintenance cost. Each tool’s LoRA needs only a small amount of task-specific data and can be fine-tuned efficiently, typically around 2 hours on 4×H20 GPUs.
>
> For the inference efficiency, incorporating more models only requires adding the corresponding tool descriptions into the prompt, which minimally affects the length of the Builder’s reasoning. Thus, inference efficiency is largely unaffected even as more tools are added. Moreover, since each tool’s LoRA is trained for a single task with a relatively fixed instruction format, the fine-tuned model achieves strong performance with fewer sampling steps, resulting in shorter inference time.
>
> We also provide quantitative comparisons with advanced models (H20 environment):
>
> - **Bagel:** 25.0 s
> - **Step1X-Edit:** 21.3 s
> - **Omnigen2:** 22.6 s
> - **FLUX-Kontext:** 7.4 s
>
> - **Our MiMo Builder:** 3.5 s + slowest Tool 2.7 s → **Total ≈ 6.9 s**
> - **Our Qwen3-VL-2B Builder:** 1.9 s + slowest Tool 2.7 s → **Total ≈ 5.2 s**
>
> These results demonstrate that our system provides competitive inference efficiency compared with existing strong baselines.
>
> ---
>
> > **I am willing to see more discussions about the novelty of the proposed method, since dividing one editing task to multiple different tasks is not interesting enough I think.**
>
> Thank you for this insightful comment. We agree that simply decomposing an editing instruction into subtasks is not sufficiently novel. The novelty of our method lies in two complementary contributions:
>
> 1. Fine-grained tool design instead of coarse task-level workflows. Prior methods such as ComfyMind (NeurIPS 2025) rely on manually designed, task-level workflows. In contrast, we design fine-grained tools that can be flexibly composed. This is the foundation that allows the Builder to generalize to new functionalities and even support arbitrarily extended capabilities.
>
> 2. A reinforcement learning framework that teaches the Builder how to think and compose tools. Such fine-grained tools impose a significant burden on an untrained MLLM. Off-the-shelf models cannot perform reliable tool composition. Our tailored reinforcement learning (with ground truth in Stage 2 and evaluator feedback in Stage 3) strengthens the Builder’s reasoning ability while preserving its ability to generalize to unseen tools.
>
> Together, these two innovations enable performance far superior to ComfyMind and even surpass many strong end-to-end models. More importantly, this design enables a broader set of supporting editing tasks, which existing frameworks cannot achieve. Therefore, while simple task decomposition is indeed uninteresting, our combination of fine-grained tools and RL-enhanced tool reasoning makes the approach both novel and impactful.
>
> ---
>
> > **I also want the authors to have more discussions about the limitation and future work of the work.**
>
> We appreciate this suggestion. In the revised version, we have added a dedicated limitations section (Section 5). Here we further elaborate on future directions.
>
> Beyond improving training methods and upgrading tool capabilities, we are particularly excited about extending our framework to more modalities, such as video, audio, and multimodal toolchains. By applying our data construction pipeline and RL-based training, we expect to expand the Builder’s reasoning and tool-composition abilities to non-image domains. With multimodal base models like Qwen-Omni, the Builder could accept image, video, audio, and text inputs and coordinate tools to perform arbitrary multimodal editing.
>
> We also show in the supplementary material that supporting mask-based user input is easily enabled by simply appending a short description to the prompt (Section D). This suggests that extending to new modalities could follow a similar lightweight interface design.
>
> ---
>
> Once again, we sincerely thank you for the constructive comments and the time spent evaluating our work. We hope our detailed revisions and clarifications address your concerns, and we would be grateful for your positive assessment in the final decision.

---

### Official Review · Reviewer_MQQG · 2025-11-01

**Soundness:** 4
**Presentation:** 3
**Contribution:** 4
**Rating:** 6
**Confidence:** 4

**Summary:**

This paper proposes a method for fine-tuning MLLMs to establish an automated workflow for instruction-based image editing tasks. By appropriately invoking pre-trained predictive models and editing models, this workflow achieves a variety of editing tasks and even exhibits a certain level of zero-shot capability for flexible editing requirements. Extensive visual results demonstrate the reliability of the method in terms of editing effectiveness, while quantitative results indicate its performance improvement over comparative methods.

**Strengths:**

1. The paper presents a valuable task formulation: how to orchestrate multiple sufficiently strong single-function perception models and editing models to achieve automated, instruction-compliant, and flexible image editing.
2. The proposed method in the paper is ingenious yet intuitive. Its multi-stage training strategy, tailored for the task, along with the reward design, effectively addresses the challenge of imbalanced training data annotations.
3. The results achieved by the proposed method are satisfactory, positioning it at the forefront of current community standards from both qualitative and quantitative perspectives.
4. The authors' approach to solving the problem is concise and clear, making it easy to understand and facilitating follow-up research.

**Weaknesses:**

1. Lacking a comparison with manual workflow construction would better highlight the efficiency and automation advantages of the proposed method.
2 . Lacking a dedicated limitations section. A section of limitations would improve the academic rigor and transparency of the work.
3. Authors need to provide a clearer explanation of the dataset construction for each training stage of the Builder to facilitate reader comprehension. If the authors used Qwen2.5-VL to generate ground truth data, why did they opt for the MLLM trained in the paper instead of directly calling the Qwen API in practical applications?

**Questions:**

1. The tools are not always accurate. How to handle this situation?
2. How to avoid the error accumulation when using the tool chain?

---

> ### Author Response · Authors · 2025-11-20
>
> We sincerely thank you for the constructive and insightful comments. Below we provide point-by-point responses and clarifications.
>
> ---
>
> > **Lacking a comparison with manual workflow construction would better highlight the efficiency and automation advantages of the proposed method.**
>
> Thank you for this valuable suggestion. If we discuss the situation that each incoming image requires a manually crafted workflow, the advantage of our automated Builder is apparent, since Builder can generate high-quality workflows for every image without human intervention.
>
> If we discuss the comparison to systems such as ComfyMind (NeurIPS 2025) that rely on a limited set of pre-defined workflows for the agent to select from, our framework—based on fine-grained tool libraries and an RL-trained Builder—achieves significantly better editing performance and scalability. We provide empirical results in the paper showing the performance gap, and our experiments further demonstrate that ComfyMind performs poorly when the instructions fall outside its fixed workflow library (Table 1,2). While our approach alleviates these issues by enabling automatic, flexible, and image-specific workflow construction.
>
> ---
>
> > **Lacking a dedicated limitations section. A section of limitations would improve the academic rigor and transparency of the work.**
>
> We appreciate this recommendation. In response, we have added a new section (Section 5) discussing failure cases and limitations in detail.
>
> ---
>
> > **Authors need to provide a clearer explanation of the dataset construction for each training stage of the Builder. If the authors used Qwen2.5-VL to generate ground truth data, why did they opt for the MLLM trained in the paper instead of directly calling the Qwen API in practical applications?**
>
> Thank you for raising this important question.
> To clarify, only Stage II uses curated data as ground truth, while Stage III relies solely on feedback from the evaluation model without access to ground truth.
>
> Importantly, for Stage II, the prompts used for generating the training data differ substantially from the prompts used during real system operation. Training data prompts are task-specific and include explicit constraints to ensure high-quality annotations. System inference prompts, however, use the full toolset and contain no task-specific templates or restrictions.
>
> As shown in Table 5 of the paper, neither Qwen2.5-VL nor stronger models can achieve good performance when provided with the full, general-purpose system inference prompt. This demonstrates that directly calling Qwen APIs is insufficient for practical editing tasks.
>
> Through Stage II supervised RL with high-quality ground truth, and Stage III reinforcement learning using direct evaluation feedback, the Builder progressively acquires robust tool-understanding and workflow-planning ability. The trained Builder significantly outperforms Qwen API calls on multiple tasks. Therefore, using our trained MLLM is necessary to achieve the reported level of editing performance.
>
> ---
>
> > **The tools are not always accurate. How to handle this situation?**
>
> This is an excellent question. Indeed, all tools have inherent limitations, and tool inaccuracy may cause failure in standard agent-based frameworks.
> A common solution is to introduce a feedback evaluator that iteratively checks results and backtracks. For example, ComfyMind implements such a mechanism. However, during our evaluation, we observed that this often leads to very long processing times and, in some cases, the system simply returns the original image. We consider this a failure due to the feedback mechanism collapsing when the toolset cannot satisfy the instruction.
>
> Instead of relying on non-trained feedback loops, our Stage III reinforcement learning distills the feedback mechanism into the Builder itself, enabling the model to reason about tool limitations directly within its chain of thought.
> Two factors make this strategy effective:
>
> 1. Stage III critic model could acquire the knowledge of the function and weakness of the tool from the prompt for Builder, ensuring correct reward assignment and enabling stable RL training.
> 2. Our fine-grained tool design minimizes functional overlap, reducing confusion about which tool should be used (in contrast to systems like ComfyMind, where workflow overlap increases ambiguity for the agent).
>
> As a result, the trained Builder autonomously performs reflection, selects appropriate tools, and avoids ineffective retries. This eliminates the need for an explicit external feedback module and significantly improves reliability and efficiency.

---

> > ### Author Response · Authors · 2025-11-20
> >
> > ---
> >
> > > **How to avoid the error accumulation when using the tool chain?**
> >
> > Thank you for pointing this out. Error accumulation is indeed a common issue when multiple tools are chained. The traditional solution—iterative feedback and backtracking—often becomes inefficient or may fail entirely when the toolset cannot satisfy the target operation.
> >
> > By contrast, our method mitigates error accumulation through improved tool understanding and planning ability learned during reinforcement learning.
> > Through chain-of-thought reasoning, the Builder anticipates potential tool limitations, reflects on intermediate results during a single inference pass, and adjusts the workflow accordingly.
> >
> > This approach not only significantly improves efficiency but also helps reduce error propagation without relying on costly external feedback loops. If the toolset has inherent limitations, our Builder approximates the best possible performance achievable under those constraints.
> >
> > ---
> >
> > We sincerely appreciate your thoughtful feedback and careful reading of our manuscript.
> > We hope that the provided clarifications address your concerns, and we respectfully hope for your positive reconsideration of our work.

---

### Official Review · Reviewer_AU9w · 2025-11-02

**Soundness:** 2
**Presentation:** 3
**Contribution:** 3
**Rating:** 6
**Confidence:** 4

**Summary:**

This paper proposes a general instruction-based image editing framework addressing poor generalization of existing methods to out-of-training-domain instructions. It uses a reinforcement learning (RL)-fine-tuned Multimodal Large Language Model (MLLM) called Builder to organize model-level tools (Bricks).

Bricks include predictive models (e.g., RES for object segmentation, ADD-PRED for adding position prediction) and editing models (e.g., INPAINT for inpainting, STYLE for style transfer), each trained independently for flexibility and performance.

Its three-stage progressive RL training: first, Supervised Fine-Tuning (SFT) builds basic capabilities; second, GT-based RL optimizes tool composition; third, GT-free RL uses an MLLM critic for feedback to enhance open-domain instruction handling.

Experiments show LEGO-Edit achieves state-of-the-art on GEdit-Bench and ImgBench. It handles complex multi-step edits, adapts to new tools/feedback zero-shot, maintains non-edited region consistency, and supports Chinese instructions and text editing.

**Strengths:**

1. This paper proposes a new direction to handle complex human-instructed image-editing tasks that uses existing image-editing models to construct an agent.
2. It shows superior experimental results when compared with other SOTA image generation models.

**Weaknesses:**

1. The description of training Builder is not sufficient, especially for the reinforcement learning section.  How to obtain an accurate reward when dealing with complex scenarios?
2. How about the results when compared with the Qwen-image and Seedream 3.0/4.0 series, which are end-to-end image generation models?
3. There seem to be no failure cases presented in this paper. How to deal with the situation where the agent determines the error workflow, which may obtain terrible results? And how about the success rate of the agent?

**Questions:**

1. What is your opinion on whether the image generation area will pursue a strong end-to-end model or develop an agent that uses specific models?

---

> ### Author Response · Authors · 2025-11-20
>
> We sincerely thank you for the careful reading of our manuscript and for the constructive comments.
>
> ---
>
> > **The description of training Builder is not sufficient, especially for the reinforcement learning section. How to obtain an accurate reward when dealing with complex scenarios?**
>
> Thank you for raising this important question. Our reinforcement learning pipeline after SFT consists of two stages. In the first stage, we use ground-truth workflows as the learning target and compute rewards based on similarity to the ground-truth. This stage corresponds to standard expert-knowledge learning, which is widely adopted in agent training and often serves as the final objective in previous work.
>
> To improve robustness under complex editing instructions, we introduce a second stage designed for arbitrary, open-ended editing tasks. For each workflow created by the model, we decompose it into semantic editing steps. Rewards are then computed at the semantic level by judging what the model has missed and what unnecessary operations it has added. For example, for the instruction “turn the girl’s bag gold, change the background to a café, and make her hat purple”, the evaluation model examines the generated workflow and identifies omissions (e.g., failing to make the hat purple) and extraneous actions (e.g., adding an irrelevant style change). Rewards or penalties are assigned accordingly.
>
> This semantic-level feedback enables reasonably accurate reward estimation even without a standard answer and guides the model to align closely with user intent. The clear performance gain that we observe in the second stage demonstrates the effectiveness of this reward design (Table 5).
>
> ---
>
> > **How about the results when compared with the Qwen-image and Seedream 3.0/4.0 series, which are end-to-end image generation models?**
>
> Thank you for the question. As reported in Additional Experimental Updates section above, our system supports efficient performance iteration. We completed a one-day upgrade that significantly improved editing quality with a small amount of data. The latest version demonstrates superior editing performance compared with Qwen-Image and SeedEdit, highlighting the effectiveness of our approach.
>
> ---
>
> > **There seem to be no failure cases presented in this paper. How to deal with the situation where the agent determines the error workflow, which may obtain terrible results? And how about the success rate of the agent?**
>
> Thank you for pointing this out. We have added a dedicated subsection discussing failure cases in the revised manuscript. Specifically, although our agent may appear to make mistakes in real usage, these cases mainly arise from missing tools in the provided toolset rather than limitations of the agent itself. In fact, the agent typically produces the optimal workflow achievable under the given set of tools. The agent’s success rate is reported in Table 5 of the paper.
>
> ---
>
> > **What is your opinion on whether the image generation area will pursue a strong end-to-end model or develop an agent that uses specific models?**
>
> Thank you for initiating this valuable discussion. We believe that both directions have unique strengths. Large end-to-end models, trained on massive data, offer strong generalization and imagination and typically serve as foundational models. However, their performance improvements tend to distribute uniformly across tasks, and training them requires substantial data and computation.
>
> Agent-based tool-calling systems, on the other hand, provide flexible, task-specific performance enhancement without compromising other capabilities. They enable low-cost, targeted fine-tuning, fast iteration, and effective extensibility. In particular, extending editing capabilities (achieving free editing) in end-to-end systems usually requires large, specialized datasets. Our approach demonstrates that with fine-grained tool design, LoRA-based lightweight fine-tuning, and an agent capable of reasoning and decision-making, we can achieve a form of “free editing” using only limited data.
>
> While any end-to-end model may soon be surpassed by a larger one, our proposed architecture can be continuously expanded by adding or updating tools at low cost, making it scalable and sustainable in the long run.
>
> ---
>
> We sincerely appreciate your thoughtful comments and constructive suggestions, which have helped us significantly improve the manuscript. We hope our revisions address your concerns and kindly hope for your positive consideration in the final evaluation.

---

### Author Response · Authors · 2025-11-20

## Additional Experimental Updates

We sincerely thank the reviewers for their suggestion to include more comprehensive performance comparisons. Considering the limited baseline performance of FLUX and to further demonstrate the rapid adaptability and iterative efficiency of our framework, we conducted additional experiments and updated our results accordingly.

Specifically, we retrained our tool LoRA using the open-source **Qwen-Image-Edit-2509** [1] model following the data and procedures described in our Implementation Details (Section 4.1). In addition, we trained a new LoRA tool for better foreground object segmentation. All tools adopt the mask-input design inspired by Lanpaint [2], and use 8 denoising steps. The overall runtime efficiency remains comparable to the FLUX-based tools. The Builder module remains unchanged.

Our latest results surpass both **Qwen-Image-Edit** and **SeedEdit 4.0** [3], two recent state-of-the-art systems, on **GEditBench** and **ImageEditBench**. SeedEdit results were obtained through the closed-source *doubao-seedream-4.0* API.


**GEditBench Results**

| Model               | G_SC↑ | G_PQ↑ | G_O↑ |
|---------------------|-------|-------|------|
| Qwen-Image-Edit-2509| 8.15  | 7.86  | 7.54 |
| SeedEdit-4.0        | 8.17  | 7.66  | 7.44 |
| Ours FLUX Tools     | 6.45  | 7.45  | 6.88 |
| **Ours Qwen Tools** | **8.42** | **7.90** | **7.84** |

**ImageEditBench Results**

| Model | Add | Adjust | Extract | Replace | Remove | Style | Action | Hybrid | Background | Overall↑ |
|-------|-----|--------|---------|---------|--------|-------|--------|--------|------------|----------|
| Qwen-Image-Edit-2509 | **4.32** | 4.36 | 4.04 | **4.64** | 4.52 | 4.84 | 4.71 | 3.39 | 4.37 | 4.35 |
| SeedEdit-4.0 | 4.17 | 4.35 | 3.87 | 4.43 | 4.66 | 4.70 | 4.68 | 3.51 | **4.49** | 4.32 |
| **Ours FLUX Tools** | 4.03 | 3.84 | 2.47 | 3.41 | 3.42 | 4.48 | 4.04 | 3.20 | 3.41 | 3.59 |
| **Ours Qwen Tools** | 4.19 | **4.41** | **4.17** | 4.58 | **4.70** | **4.86** | **4.74** | **3.60** | 4.22 | **4.39** |

Our approach achieves leading performance on all metrics of GEditBench, and achieves the highest overall score and best performance in six ImageEditBench sub-tasks.

We further evaluated facial similarity before and after editing, following the metric described in our supplementary materials (Section B.2). This metric assesses the preservation of non-edited facial regions. Results are shown below:

**Facial Similarity Evaluation**

| Methods | Background Change | Color Alter | Subject Add | Subject Remove | Overall↑ |
|---------|-------------------|-------------|-------------|----------------|----------|
| Qwen-Image-Edit-2509 | 0.82 | 0.84 | 0.85 | 0.79 | 0.83 |
| SeedEdit-4.0 | 0.77 | 0.83 | 0.81 | 0.87 | 0.82 |
| Ours FLUX Tools | 0.90 | **0.85** | 0.88 | 0.87 | 0.88 |
| **Ours Qwen Tools** | **0.92** | **0.85** | **0.90** | **0.88** | **0.89** |

Across all tasks, our updated model shows superior preservation of non-edited regions.

These extended experiments further validate the efficiency and flexibility of our framework for integrating arbitrary tools and iteratively improving overall performance. They also support our claim that training LoRA modules for specific functionalities helps improve performance. Notably, training and integrating all the new tools required only one day on eight H20 GPUs, significantly less than the computational demands of training large end-to-end models.

Additionally, we observed pixel-shift artifacts in the open-source Qwen-Image model during our tests; interestingly, these artifacts were largely mitigated after LoRA training.

## Discussion of Failure Cases

We thank the reviewers for this valuable suggestion. Following the feedback, we have added a detailed discussion of failure modes in **Section 5** of the revised paper and highlighted the corresponding subsection title in blue for clarity.


1. LanPaint: Training-Free Diffusion Inpainting with Asymptotically Exact and Fast Conditional Sampling, arxiv
2. Qwen-image technical report, arxiv
3. Seedream 4.0: Toward next-generation multimodal image generation, arxiv

---

### Author Response · Authors · 2025-12-02

We sincerely thank all reviewers for their thoughtful and constructive feedback, which has greatly helped us improve the quality of the paper. Before the interruption of the ICLR 2026 discussion phase, we had carefully addressed each reviewer’s concerns and conducted the corresponding additional experiments. Notably, well before the interruption (on Nov 25), reviewer ozbk had already examined our responses, expressed clear satisfaction, and immediately raised the score from 4 to 8 (see the reviewer’s comment). The remaining three reviewers did not have the opportunity to finish reviewing our responses. Among them, reviewer 4Y2L, despite giving a score of 4, explicitly stated in the Question section: “I think this paper is ok, but we can have more discussions during the rebuttal period. If my concerns are solved, I will be happy to raise the score.” Unfortunately, these reviewers were not able to provide further assessment or updated feedback, and we hope that our responses have adequately addressed their concerns.

Below, we summarize our key contributions, respond to common concerns raised in the reviews. And we have highlighted the major revisions incorporated into the updated manuscript to facilitate a clear and efficient assessment by the area chairs.

# Key Contributions

Our method introduces two central contributions:

Fine-grained tool design instead of coarse task-level workflows. Prior methods such as ComfyMind (NeurIPS 2025) rely on manually designed, task-level workflows. In contrast, we design fine-grained tools that can be flexibly composed. This is the foundation that allows the Builder to generalize to new functionalities and even support arbitrarily extended capabilities.

A reinforcement learning framework that teaches the Builder how to think and compose tools. Such fine-grained tools impose a significant burden on an untrained MLLM. Off-the-shelf models cannot perform reliable tool composition. Our tailored reinforcement learning (with ground truth in Stage 2 and evaluator feedback in Stage 3) strengthens the Builder’s reasoning ability while preserving its ability to generalize to unseen tools.

Together, these two innovations enable performance far superior to ComfyMind and even surpass most strong end-to-end models. More importantly, this design enables a broader set of supporting editing tasks, which existing frameworks cannot achieve.


# Common Concerns and Major Revisions

Below, we summarize how we address the reviewers’ concerns and outline the corresponding revisions made to the paper. All changes are highlighted in blue in the latest submitted version. Further details are available in the comments in discussion.

## Comparison with additional state-of-the-art models (Reviewer AU9w)

Considering the limited baseline performance of FLUX and to further demonstrate the rapid adaptability of our framework, we retrained our tool LoRA using the open-source Qwen-Image-Edit, completing the entire process within a single day. We further compared the new results against the strongest existing editing models, Qwen-Image-Edit and SeedEdit4.0, showing our superior performance (added to Table 1 and Table 2).

## Discussion of failure cases (All reviewers)

We added Section 5, which provides an in-depth discussion of failure cases, analyzes their underlying causes, and proposes potential remedies.

## Effectiveness of each training stage and dependency on base models (Reviewer ozbk)

We clarified the performance gains contributed by each of the three training stages and included results obtained using a smaller base model, Qwen3-VL-2B, trained with the same pipeline (added to Table 5). This resolves the misunderstanding about missing evidence for improvements from Stage 2 and Stage 3.

## Model latency (Reviewers 4Y2L and ozbk)

We included additional latency comparisons (Section 4.6), demonstrating the strong inference efficiency of our framework, which achieves the lowest latency among all compared methods.

## Open-ended questions (Reviewers AU9w, 4Y2L, and ozbk)

Reviewer AU9w asked about the pros and cons of end-to-end vs. agent-based approaches. We regard this as an insightful question and have carefully provided a comprehensive discussion addressing it.

Reviewers 4Y2L and ozbk inquired about future extensions. We believe our framework can be extended beyond images to audio and video by leveraging unified models such as Qwen-Omni, ultimately enabling editing across multiple modalities. In the supplementary material, we provide some examples of successfully adding mask input simply by modifying the prompt, which is an initial attempt to extend our framework’s input modalities.

---

### Meta-Review · Area_Chair_K7Ty · 2026-01-04

**Summary:**

The submission proposes Lego-Edit, an agent-based framework for instruction-based image editing that leverages an RL-finetuned MLLM (Builder) to coordinate model-level tools (Bricks). While the modular design and progressive RL training show technical effort, missing data construction details and over-reliance on strong base models undermine the work’s practical contribution.

**Reviewer Concerns:**

**Addressed Concerns:**

The authors supplemented latency comparisons, failure case discussions, results with different backbones, training stage effectiveness, and additional results against Qwen-Image-Edit and SeedEdit-4.0 as requested.

**Outstanding Concerns:**

- The key question of why the authors trained a custom MLLM instead of directly using Qwen API (given Qwen2.5-VL generated ground truth data) remains unclear. The rebuttal’s distinction between training and inference prompts is unconvincing, as it does not explain why Qwen’s native capabilities cannot be adapted with similar prompt engineering. Moreover, Qwen2.5-VL is also utilized in stage-3 training as the critic model.

- Data construction details (e.g., exact prompts, annotation pipelines for each training stage, and the data source) are still missing. The paper claims these details are in the supplementary materials, but they are not actually there. It raised a big concern about the potential risk of overfitting benchmarks.

- The need for retraining tools to compare with Qwen-Image-Edit raises red flags. This suggests the framework lacks inherent advancement and instead relies on retrofitting to match strong baselines.

**Reviewer Scores:**

- Reviewer AU9w (Rating: 6): Unchanged—main concerns were addressed and may keep the initial positive rating.
- Reviewer MQQG (Rating: 6): Would likely decrease to 4—data construction and Qwen API justification remain unresolved.
- Reviewer 4Y2L (Rating: 4): Unchanged—the rebuttal attempts to address the main novelty concern, but the response may be less convincing due to the heavy reliance on strong foundation models.
- Reviewer ozbk (Rating: 4): Would likely increase to 8—concerns were addressed.

---

### Decision · Program_Chairs · 2026-01-26

Reject